# Efficient constrained sampling via the mirror-Langevin algorithm

**Kwangjun Ahn**
Department of EECS
Massachusetts Institute of Technology
Cambridge, MA 02139
kjahn@mit.edu

**Sinho Chewi**
Department of Mathematics
Massachusetts Institute of Technology
Cambridge, MA 02139
schewi@mit.edu

## Abstract

We propose a new discretization of the mirror-Langevin diffusion and give a crisp proof of its convergence. Our analysis uses relative convexity/smoothness and self-concordance, ideas which originated in convex optimization, together with a new result in optimal transport that generalizes the displacement convexity of the entropy. Unlike prior works, our result both (1) requires much weaker assumptions on the mirror map and the target distribution, and (2) has vanishing bias as the step size tends to zero. In particular, for the task of sampling from a log-concave distribution supported on a compact set, our theoretical results are significantly better than the existing guarantees.

## 1 Introduction

We consider the following canonical sampling problem. Let $V : \mathbb{R}^d \to \mathbb{R} \cup \{\infty\}$ be a convex function and let $\pi$ be the density on $\mathbb{R}^d$ which is proportional to $\exp(-V)$. The task is to output a sample which is (approximately) distributed according to $\pi$, given query access to the gradients of $V$.

The sampling problem has attracted considerable attention recently within the machine learning and statistics communities. This renewed interest in sampling is spurred, on one hand, by a wide breadth of applications ranging from Bayesian inference [RC04, DM+19] and its use in inverse problems [DS17], to neural networks [GPAM+14, TR20]. On the other hand, there is a deep and fruitful connection between sampling and the field of *optimization*, introduced in the seminal work [JKO98], which has resulted in the rapid development of sampling algorithms inspired by optimization methods such as: proximal/splitting methods [Ber18, Wib18, Wib19, SKL20], coordinate descent [DLLW21a, DLLW21b], mirror descent [HKRC18, CLGL+20, ZPFP20], Nesterov's accelerated gradient descent [CCBJ18, MCC+21, DRD20], and Newton methods [MWBG12, SBCR16, CLGL+20, WL20].

To describe this connection, we recall the *Langevin diffusion*, which is the solution to the following stochastic differential equation (SDE):

$$\mathrm{d}X_t = -\nabla V(X_t)\,\mathrm{d}t + \sqrt{2}\,\mathrm{d}B_t. \tag{LD}$$

Under standard assumptions on the potential $V$, the SDE is well-posed and it converges in distribution, as $t \to \infty$, to its unique stationary distribution $\pi$. Thus, once suitably discretized, it yields a popular algorithm for the sampling problem. The Langevin diffusion is classically studied using techniques from Markov semigroup theory [see, e.g. BGL14, Pav14], but there is a more insightful perspective which views the diffusion (LD) through the lens of optimization [JKO98]. Specifically, if $\mu_t$ denotes the law of the process (LD) at time $t$, then the curve $(\mu_t)_{t\geq 0}$ is the *gradient flow* of the KL divergence $\mathcal{D}_{\mathsf{KL}}(\cdot \parallel \pi)$ in the Wasserstein space of probability measures. This perspective has not only inspired

35th Conference on Neural Information Processing Systems (NeurIPS 2021).

Figure 1: Illustration of the mirror Langevin algorithm (MLA). This illustration is adapted from [Bub15, Figure 4.1].

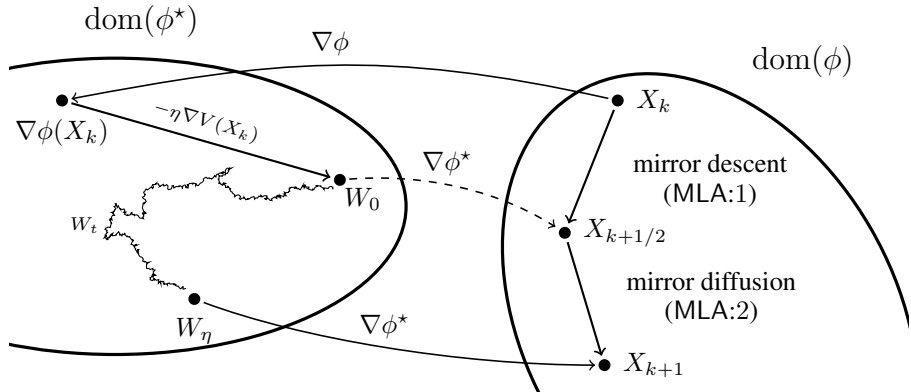

new analyses of Langevin [CB18, Wib18, DMM19, VW19], but has also emboldened the possibility of bringing to bear the extensive toolkit of optimization onto the problem of sampling; see the references listed above.

However, the vanilla Langevin diffusion notably fails when the support of the target distribution $\pi$ is not all of $\mathbb{R}^d$. This task of *constrained sampling*, named in analogy to constrained optimization, arises in applications such as latent Dirichlet allocation [BNJ03], ordinal data models [JA06], survival time analysis [KM06], regularized regression [CEAM+12], and Bayesian matrix factorization [PBJ14]. Despite such a broad range of applications, the constrained sampling problem has proven to be challenging. In particular, most prior works have focused on domain-specific algorithms [GSL92, PP14, LS16], and the first general-purpose algorithms for this task are recent [BDMP17, BEL18].

In this work, we tackle the constrained sampling problem via the *mirror-Langevin algorithm* (MLA). MLA is a discretization of the mirror-Langevin diffusion [HKRC18, ZPFP20], which is the sampling analogue of *mirror descent*. Namely, if $\phi : \mathbb{R}^d \to \mathbb{R} \cup \{\infty\}$ is a mirror map, then the mirror-Langevin diffusion is the solution to the SDE

$$X_t = \nabla\phi^\star(Y_t), \qquad \mathrm{d}Y_t = -\nabla V(X_t)\,\mathrm{d}t + \sqrt{2}\,[\nabla^2\phi(X_t)]^{1/2}\,\mathrm{d}B_t\,. \qquad \text{(MLD)}$$

**Technical motivation.** Recently, Zhang et al. [ZPFP20] analyze an Euler-Maruyama discretization of MLD; see (2.2) for details. The most curious aspect of their result is that their convergence guarantee has a bias term that does not vanish even when step size tends to zero and the number of iterations tends to infinity. Moreover, they conjecture that this bias term is unavoidable. This is in contrast to known results for standard Langevin, which raises the main question of this paper:

*Can a different discretization of* MLD *lead to a vanishing bias?*

**Our contributions**. We propose a new discretization of the mirror-Langevin diffusion, given in (MLA) and illustrated in Figure 1. Our proposed discretization has the same cost as the standard Euler-Maruyama discretization of MLD in terms of the number of queries to the gradient oracle for $V$. We remark that our scheme for the case $\phi = \|\cdot\|^2/2$ recovers the unadjusted Langevin algorithm. The most important aspect of our result is that the bias of our algorithm vanishes as the step size tends to zero unlike the result by Zhang et al. [ZPFP20].

By adapting the analysis of Durmus et al. [DMM19], we provide a clean convergence analysis of our algorithm which theoretically validates our discretization scheme. Notably, our analysis only requires standard assumptions/definitions which are well-studied in optimization. In particular, we establish a stronger link between sampling and optimization without relying on technical assumptions of Zhang et al. [ZPFP20] (e.g. commutation conditions for Hessians; see (A5) therein).

Moreover, our analysis combines ideas from optimization with the calculus of optimal transport. In particular, we establish a new generalization of a celebrated fact, namely that the entropy functional is

*displacement convex* along Wasserstein geodesics, to the setting of Bregman divergences (Theorem 4). This inequality has interesting consequences in its own right; as we discuss in Corollary 1, our result already implies the transport inequality of Cordero-Erausquin [CE17].

We provide convergence guarantees for the following classes of potentials: (1) convex and relatively smooth (Theorem 1); (2) strongly relatively convex and relatively smooth (Theorem 2); and (3) convex and Lipschitz (Theorem 3). Our results largely match state-of-the-art results for the discretization of the Langevin algorithm for unconstrained sampling. Our work paves the way for the practical deployment of mirror-Langevin methods for sampling applications, paralleling the successes of mirror descent in optimization [NY83, JN11, Bub15].

In Section 5, we demonstrate the strength of our convergence guarantees compared with the previous works [BDMP17, BEL18] in an application to Bayesian logistic regression; further applications are given in Appendix E. We also corroborate our theoretical findings with numerical experiments.

**Other related works.** Recently, a few works have proposed modifications of the Langevin algorithm for the task of constrained sampling. Bubeck et al. [BEL18] studied the *projected Langevin algorithm* (PLA), which simply projects each step of the Langevin algorithm onto $\mathrm{dom}(V)$. A different approach was taken in Brosse et al. [BDMP17], which applies the Langevin algorithm to a smooth approximation of $V$ given by the Moreau-Yosida envelope. The latter approach was later interpreted and further analyzed by Salim and Richtarik [SR20] using the primal-dual optimality framework from convex optimization.

A different line of work, more closely related to ours, uses a mirror map to change the *geometry* of the sampling problem [HKRC18, CLGL$^+$20, ZPFP20]. In particular, the mirror-Langevin diffusion (MLD) was first introduced in an earlier draft of [HKRC18], as well as in [ZPFP20]. The diffusion was further studied in [CLGL$^+$20], which provided a simple convergence analysis in continuous time using the sampling analog of *Polyak-Łojasiewicz inequalities* [KNS16]. We also remark that the idea of changing the geometry via a mirror map also played an crucial role for the problem of sampling from the uniform distribution over a polytope [KN12, LV17, CDWY18, LV18, GN20, LLV20].

Lastly, our work follows the trend of applying ideas from optimization to the task of sampling. Specifically, our analysis adopts the framework of relative convexity and smoothness, which was advocated as a more flexible framework for optimization in [BBT17, LFN18].

## 2 The mirror-Langevin algorithm

### 2.1 Background

In this section, we list basic definitions and assumptions that we employ in this work.

**Convex functions of Legendre type**. Throughout, we assume familiarity with the basic notions of convex analysis [see e.g. Roc70, BL06].

**Definition 1** (Convex functions of Legendre type [Roc70, §26])**.** *A proper convex lower semicontinuous function* $\phi : \mathbb{R}^d \to \mathbb{R} \cup \{\infty\}$ *is* of Legendre type *if*

   *(i)* $\mathcal{Q} := \mathrm{int}(\mathrm{dom}(\phi)) \neq \emptyset$,

   *(ii)* $\phi$ *is strictly convex and differentiable on* $\mathcal{Q}$, *and*

   *(iii)* $\lim_{k \to \infty} \|\nabla\phi(x_k)\| = \infty$ *whenever* $\{x_k\}_{k \in \mathbb{N}}$ *is a sequence in* $\mathcal{Q}$ *converging to* $\partial\mathcal{Q}$.

The key properties of convex functions of Legendre type are listed below:

- The subdifferential $\partial\phi$ is single-valued and hence $\partial\phi = \{\nabla\phi\}$ [Roc70, Theorem 26.1].
- $\phi$ is a convex function of Legendre type if and only if its Fenchel conjugate $\phi^\star$ is a convex function of Legendre type [Roc70, Theorem 26.5].
- The gradient $\nabla\phi$ forms a bijection between $\mathrm{int}(\mathrm{dom}(\phi))$ and $\mathrm{int}(\mathrm{dom}(\phi^\star))$ with $\nabla\phi^\star = (\nabla\phi)^{-1}$ [Roc70, Theorem 26.5].

We refer readers to [Roc70, §26] for more details. We henceforth assume that our mirror map $\phi$ is a convex function of Legendre type.

The natural notion of "distance" associated with the mirror map $\phi$ is given by the Bregman divergence [see, e.g. Bub15, §4]:

**Definition 2** (Bregman divergence [Bre67]). *For a convex function $\phi$ of Legendre type, the Bregman divergence $D_\phi(\cdot, \cdot)$ associated to $\phi$ is defined as*

$$D_\phi(x, y) := \phi(x) - \phi(y) - \langle \nabla \phi(y), x - y \rangle, \qquad \forall x, y \in \mathcal{Q} := \text{int}(\text{dom}(\phi)).$$

The Bregman divergence behaves like a squared distance; indeed, as $x \to y$ a Taylor expansion shows that $D_\phi(x, y) \sim \frac{1}{2} \langle x - y, \nabla^2 \phi(y)(x - y) \rangle$. We refer to [Bub15, §4] for other basic properties of the Bregman divergence. Hereinafter, when we use a Bregman divergence, we implicitly assume that its associated function $\phi$ is a mirror map.

An important case to keep in mind is the mirror map $\phi = \frac{\|\cdot\|^2}{2}$, where $\|\cdot\|$ denotes the Euclidean norm, in which case the Bregman divergence simply becomes $D_\phi(x, y) = \frac{1}{2} \|x - y\|^2$.

**Self-concordance**. We recall the definition of self-concordance, which has been extensively used in applications such as interior-point methods [NN94]. Given a $C^2$ strictly convex function $\phi$, the local norm at $x \in \text{int}(\text{dom}(\phi))$ with respect to $\phi$ is defined as

$$\|u\|_{\nabla^2 \phi(x)} = \sqrt{\langle \nabla^2 \phi(x) u, u \rangle} \qquad \text{for all } u \in \mathbb{R}^d.$$

The dual local norm at $x \in \text{int}(\text{dom}(\phi))$ with respect to $\phi$ is

$$\|u\|_{[\nabla^2 \phi(x)]^{-1}} = \sqrt{\langle [\nabla^2 \phi(x)]^{-1} u, u \rangle} \qquad \text{for all } u \in \mathbb{R}^d.$$

**Definition 3** (Self-concordant function [Nes18, §5.1.3]). *We say that a $C^3$ convex function $\phi$ is self-concordant with a constant $M_\phi \geq 0$ if for any $x \in \text{int}(\text{dom}(\phi))$*

$$|\nabla^3 \phi(x)[u, u, u]| \leq 2M_\phi \|u\|_{\nabla^2 \phi(x)}^3 \qquad \text{for all } u \in \mathbb{R}^d.$$

**Relative convexity/smoothness**. We recall the following definitions:

**Definition 4** (Relative convexity [BBT17, LFN18]). *$V$ is $\alpha$-convex relative to $\phi$ if*

$$V(y) \geq V(x) + \langle \nabla V(x), y - x \rangle + \alpha D_\phi(y, x) \qquad \forall x, y \in \mathcal{Q}.$$

**Definition 5** (Relative smoothness [BBT17, LFN18]). *$V$ is $\beta$-smooth relative to $\phi$ if*

$$V(y) \leq V(x) + \langle \nabla V(x), y - x \rangle + \beta D_\phi(y, x) \qquad \forall x, y \in \mathcal{Q}.$$

We give some basic facts about these definitions in Appendix B.

In the rest of this work, we assume that $V \in C^2(\mathcal{X})$ where $\mathcal{X} := \text{int}(\text{dom}(V))$, that $\mathcal{X} \subseteq \overline{\mathcal{Q}}$, and $\mathcal{X} \cap \mathcal{Q} \neq \emptyset$. Also, we assume that $\exp(-V)$ is integrable so that $\pi$ is well-defined; this holds if and only if $V(x) \geq a \|x\| - b$ for some $a, b > 0$ [BGVV14, Lemma 2.2.1].

**Optimal transport**. Given a lower semicontinuous cost function $c : \mathbb{R}^d \times \mathbb{R}^d \to [0, \infty]$, we can define the *optimal transport* cost between two probability measures $\mu$ and $\nu$ on $\mathbb{R}^d$ to be

$$\inf\{\mathbb{E}\, c(X, Y) \mid X \sim \mu,\ Y \sim \nu\}. \tag{2.1}$$

Here, the infimum is taken over pairs of random variables $(X, Y)$ defined on the same probability space, with marginal laws $\mu$ and $\nu$ respectively. It is known that the infimum in (2.1) is always attained; we refer to the standard introductory texts [Vil03, Vil09, San15] for this and other basic facts in optimal transport.

In this work, we are most concerned with the case when the cost function $c$ is the Bregman divergence associated with a mirror map:

**Definition 6** (Bregman transport cost). *The Bregman transport cost is defined as*

$$\mathcal{D}_\phi(\mu, \nu) := \inf\{\mathbb{E}\, D_\phi(X, Y) \mid X \sim \mu,\ Y \sim \nu\}.$$

The Bregman transport cost was also studied in [CE17].

In particular, when $\phi = \frac{\|\cdot\|^2}{2}$, we obtain an important special case:

**Definition 7** (2-Wasserstein distance). *The 2-Wasserstein distance $W_2$ is defined as*

$$W_2^2(\mu, \nu) := \inf\{\mathbb{E}[\|X - Y\|^2] \mid X \sim \mu, \ Y \sim \nu\}.$$

The $W_2$ optimal transport cost indeed defines a metric over the space of probability measures on $\mathbb{R}^d$ with finite second moment [Vil03, Theorem 7.3]; we refer to this metric space as the *Wasserstein space*. The $W_2$ metric is particularly important because it arises from a formal Riemannian structure on the Wasserstein space. This perspective was introduced in [Ott01] and applied to the Langevin diffusion in [JKO98, OV00]; in particular, these latter two works justify the perspective of the Langevin diffusion as a gradient flow of the Kullback-Leibler divergence in Wasserstein space. A rigorous exposition to Wasserstein calculus can be found in [AGS08, Vil09].

Here, we give a brief and informal introduction to the calculation rules of optimal transport. For any regular curve of measures $(\mu_t)_{t \geq 0}$, there is a corresponding family of *tangent vectors* $(v_t)_{t \geq 0}$ [see AGS08, Theorem 8.3.1]; here, $v_t : \mathbb{R}^d \to \mathbb{R}^d$ is a vector field on $\mathbb{R}^d$. Also, if $\mathcal{F}$ is any well-behaved functional defined over Wasserstein space, then at each regular measure $\mu$ one can define the *Wasserstein gradient* of $\mathcal{F}$ at $\mu$, which we denote $\nabla_{W_2} \mathcal{F}(\mu)$; it is also a mapping $\mathbb{R}^d \to \mathbb{R}^d$. Then, we have the calculation rule

$$\partial_t \mathcal{F}(\mu_t) = \mathbb{E}\langle \nabla_{W_2} \mathcal{F}(\mu_t)(X_t), v_t(X_t)\rangle$$

for any regular curve of measures $(\mu_t)_{t \geq 0}$ with corresponding tangent vectors $(v_t)_{t \geq 0}$, where $X_t \sim \mu_t$. We will use this calculation rule in Appendix C.

## 2.2 Discretization of the mirror-Langevin diffusion

In order to turn a continuous-time diffusion such as (MLD) into an implementable algorithm, it is necessary to first discretize the stochastic process. The discretization considered in the prior works [ZPFP20, CLGL$^+$20] is a simple Euler-Maruyama discretization: fixing $\eta > 0$, we define a sequence of iterates $(X_k)_{k \in \mathbb{N}}$ via

$$\nabla\phi(X_{k+1}) = \nabla\phi(X_k) - \eta \nabla V(X_k) + \sqrt{2\eta}\left[\nabla^2\phi(X_k)\right]^{1/2} \xi_k, \tag{2.2}$$

where $(\xi_k)_{k \in \mathbb{N}}$ is a sequence of i.i.d. standard Gaussians in $\mathbb{R}^d$.

However, many other discretizations are possible. Indeed, in many machine learning applications, the most costly step is the evaluation of $\nabla V$, which may require a sum over a large training set, whereas the mirror map $\phi$ may be chosen to have a simple form. For the purpose of obtaining a more efficient sampling algorithm, it may therefore be a favorable trade-off to use a high-precision implementation of the diffusion step at the cost of additional computation time (which nonetheless does not require additional query access to the gradients of $V$). Motivated by these considerations, we propose a new discretization (see Figure 1 for an illustration):

---

**The mirror-Langevin algorithm** (MLA)**:**

$$X_{k+1/2} := \underset{x \in \mathcal{Q}}{\arg\min}\left[\langle \eta\nabla V(X_k), x\rangle + D_\phi(x, X_k)\right], \tag{MLA:1}$$

$$X_{k+1} := \nabla\phi^\star(W_\eta), \qquad \text{where } \begin{cases} dW_t = \sqrt{2}\left[\nabla^2\phi^\star(W_t)\right]^{-1/2} dB_t, \\ W_0 = \nabla\phi(X_{k+1/2}). \end{cases} \tag{MLA:2}$$

---

In MLA:2, the stochastic processes $(W_t)_{t \geq 0}$ are assumed to be driven by independent Brownian motions at each iteration. When $\phi = \|\cdot\|^2/2$, MLA recovers the unadjusted Langevin algorithm.

**Practicality.** Although MLA:2 is defined using the exact solution of an SDE (and we analyze the exact step MLA:2 for simplicity), it should be understood as capturing the idea of discretizing the diffusion step more finely (e.g., through multiple inner iterations of an Euler-Maruyama discretization) than the gradient step. This is indeed amenable to practical implementation since, as previously discussed, the gradient step is typically much more costly than the diffusion step. Moreover, this is justified by our theoretical results in the next section, which together with the conjecture of [ZPFP20] suggest that fine discretization of MLA:2 is potentially crucial for attaining vanishing bias. Nevertheless, it is indeed the case that a single iteration of MLA is more costly than a single step of the Euler-Maruyama discretization of MLD, and this represents a limitation of our work.

**Remark 1.** Our proposed discretization can be understood as a more faithful discretization of the mirror-Langevin diffusion (MLD), *à la* [GWS21]. It can also be understood as the forward-flow discretization of MLD in the interpretation of [Wib18].

**Remark 2.** In order for MLA:2 to be well-defined, we require assumptions on $\phi$ such that the diffusion $(W_t)_{t \geq 0}$ is non-explosive, i.e., it does not exit $\text{int}(\text{dom}(\phi^\star))$ in finite time. This holds under very mild assumptions on $\phi$; see [GK96]. For situations of interest, the assumptions of [GK96] can be checked directly.

# 3 Convergence rates of mirror-Langevin algorithm

First, we state the main assumptions which are used for our main results.

**Assumption 1** (Self-concordance of $\phi$). *We assume that $\phi$ is $M_\phi$-self-concordant (Definition 3).*

**Assumption 2** (Relative Lipschitzness). *We assume that $V$ is $L$-relatively Lipschitz with respect to $\phi$, in the sense that $\|\nabla V(x)\|_{[\nabla^2 \phi(x)]^{-1}} \leq L$ for all $x \in \mathcal{X}$ (see Section 2.1 for the definition of the local norm used here).*

**Assumption 3** (Relative convexity and smoothness). *We assume that $V$ is $\alpha$-convex relative to $\phi$ and $\beta$-smooth relative to $\phi$, where $0 \leq \alpha \leq \beta \leq \infty$ (Definitions 4 and 5).*

**Remark 3.** Our analysis works under weaker assumptions than those of [ZPFP20]. In particular, our analysis does not assume the moment condition on the Hessian ((A2) therein) and the bound on the commutator between $\nabla^2 \phi$ and $\nabla^2 V$ ((A5) therein). Moreover, our analysis uses weaker (and more standard) definitions of self-concordance.

Throughout this section, we assume the conditions listed above, and we present convergence results for MLA under various sets of assumptions. Our first two results pertain to the smooth case, i.e. $\beta < \infty$. Define the parameter

$$\boxed{\beta' := \beta + 2M_\phi L.}$$

One might wonder how large $\beta'$ is for typical applications. First, we only need $\phi$ to be self-concordant, *not* a self-concordant *barrier*. Hence the appearance of $M_\phi$ is typically not problematic; for instance, a log barrier with $m$ constraints is $O(1)$-self-concordant. The smoothness parameter could be large and dimension-dependent in general. However, such situations are actually where our approach could be potentially advantageous. In Appendix E.2, we demonstrate an example where the smoothness parameter becomes much smaller by choosing $\phi$ carefully.

**Theorem 1** (Weakly convex case). *Suppose that Assumptions 1, 2, 3 hold with $\alpha = 0$ and $\beta' > 0$. For a target accuracy $\epsilon > 0$, let $X_k \sim \mu_k$ denote the iterates of MLA with step size $\eta = \min\{\frac{\epsilon}{2\beta' d}, \frac{1}{\beta'}\}$. Then, the following convergence rate holds for the mixture distribution $\overline{\mu}_N := \frac{1}{N} \sum_{k=1}^{N} \mu_k$:*

$$\mathcal{D}_{\mathsf{KL}}(\overline{\mu}_N \,\|\, \pi) \leq \epsilon, \quad \text{provided that } N \geq \frac{4\beta' d \, \mathcal{D}_\phi(\pi, \mu_0)}{\epsilon^2} \max\left\{1, \frac{\epsilon}{2d}\right\}. \tag{3.1}$$

*Proof.* See Appendix C.2. □

**Theorem 2** (Strongly relatively convex case). *Suppose that Assumptions 1, 2, 3 hold with $\alpha, \beta' > 0$.*

1. *(Convergence in Bregman transport cost) For a target accuracy $\epsilon > 0$, let $X_k \sim \mu_k$ denote the iterates of MLA with step size $\eta = \min\{\frac{\alpha\epsilon}{2\beta' d}, \frac{1}{\beta'}\}$. Then,*

$$\mathcal{D}_\phi(\pi, \mu_N) \leq \epsilon, \quad \text{provided that } N \geq \frac{2\beta' d}{\alpha^2 \epsilon} \ln\left(\frac{2\mathcal{D}_\phi(\pi, \mu_0)}{\epsilon}\right) \max\left\{1, \frac{\alpha\epsilon}{2d}\right\}.$$

2. *(Convergence in KL divergence) For a target accuracy $\epsilon > 0$, suppose that $X_0 \sim \mu_0$ satisfies $\mathcal{D}_\phi(\pi, \mu_0) \leq \epsilon/\alpha$. Let $X_k \sim \mu_k$ denote the iterates of MLA with step size $\eta = \min\{\frac{\alpha\epsilon}{2\beta' d}, \frac{1}{\beta'}\}$. Then, the following convergence rate holds for the mixture distribution $\overline{\mu}_N := \frac{1}{N} \sum_{k=1}^{N} \mu_k$,*

$$\mathcal{D}_{\mathsf{KL}}(\overline{\mu}_N \,\|\, \pi) \leq \epsilon, \quad \text{provided that } N \geq \frac{4\beta' d}{\alpha\epsilon} \max\left\{1, \frac{\epsilon}{2d}\right\}.$$

*Proof.* See Appendix C.3. □

Note that the initialization assumption $\mathcal{D}_\phi(\pi, \mu_0) \leq \epsilon/\alpha$ in the second assertion of Theorem 2 can be obtained from the first guarantee of Theorem 2. Chaining together the two parts of the theorem, we therefore obtain the following guarantee: suppose that we initialize MLA at a distribution $\mu_0$. Then, with step size $\eta = \min\{\frac{\alpha\epsilon}{2\beta'd}, \frac{1}{\beta'}\}$, we obtain

$$\mathcal{D}_{\mathsf{KL}}\Big(\frac{1}{N_1}\sum_{k=N_0+1}^{N_0+N_1}\mu_k \;\Big\|\; \pi\Big) \leq \epsilon\,, \qquad \text{provided that } \begin{cases} N_0 \geq \frac{2\beta'd}{\alpha\epsilon}\ln\big(\frac{2\alpha\mathcal{D}_\phi(\pi,\mu_0)}{\epsilon}\big)\max\{1, \frac{\epsilon}{2d}\}\,, \\ N_1 \geq \frac{4\beta'd}{\alpha\epsilon}\max\{1, \frac{\epsilon}{2d}\}\,. \end{cases}$$

Observe also that for the case $\phi = \frac{\|\cdot\|^2}{2}$, Theorems 1 and 2 recover the corresponding convergence guarantees for the unadjusted Langevin algorithm [Corollary 7, and Corollaries 10 and 11 respectively in DMM19].[1]

Next, we present our guarantee for the non-smooth case $\beta = \infty$. For this result, we assume that $\phi$ is strongly convex w.r.t. a norm $\|\cdot\|$ on $\mathbb{R}^d$, and that $V$ is $\tilde{L}$-Lipschitz in this norm. Since the norm of the gradient should be measured in the dual norm $\|\cdot\|_\star$, this means precisely that

$$\|\nabla V(x)\|_\star \leq \tilde{L}\,, \qquad \text{for all } x \in \mathcal{X}\,. \tag{3.2}$$

We also note that the next result does not require self-concordance of $\phi$.

**Theorem 3** (Non-smooth case). *Assume $\phi$ is 1-strongly convex w.r.t. a norm $\|\cdot\|$ on $\mathbb{R}^d$, that $V$ is $\tilde{L}$-Lipschitz in this norm (in the sense of (3.2)), and that $\alpha = 0$ (i.e., $V$ is convex). For a target accuracy $\epsilon > 0$, let $X_k \sim \mu_k$ denote the iterates of $\mathsf{MLA}$ with step size $\eta = \epsilon/\tilde{L}^2$. Then, the following convergence rate holds for the mixture distribution $\overline{\mu}_N := \frac{1}{N}\sum_{k=1}^N \mu_k$:*

$$\mathcal{D}_{\mathsf{KL}}(\overline{\mu}_N \,\|\, \pi) \leq \epsilon\,, \quad \text{provided that } N \geq \frac{2\tilde{L}^2\mathcal{D}_\phi(\pi, \mu_{1/2})}{\epsilon^2}\,.$$

*Proof.* See Appendix C.4. □

The assumption (3.2) is stronger than relative Lipschitzness: if $V$ satisfies (3.2) and $\phi$ is 1-strongly convex w.r.t. $\|\cdot\|$, then $V$ is $L$-relatively Lipschitz with respect to $\phi$ with $L \leq \tilde{L}$. When $\|\cdot\|$ is the Euclidean norm and $\phi = \frac{\|\cdot\|^2}{2}$, then we recover a special case of [DMM19, Corollary 14].

We now make a number of remarks about our result.

**Remark 4** (Implementing $\overline{\mu}_N$). One can output a sample from $\overline{\mu}_N$ by simply outputting one of the iterates $\{X_k\}_{k=1}^N$ chosen uniformly at random.

**Remark 5** (Convergence in other metrics). Using standard inequalities, our results for convergence in KL divergence imply convergence in a number of other information divergences such as the total variation distance, see [Tsy09, §2.4].

When $V$ is $\alpha$-strongly convex (w.r.t. $\frac{\|\cdot\|^2}{2}$), then the $T_2$ transportation-cost inequality [Vil09, Theorem 22.14] $\alpha W_2^2(\mu, \pi) \leq \mathcal{D}_{\mathsf{KL}}(\mu \,\|\, \pi)$ implies convergence in the $W_2$ distance as well. In general, we do not have convergence in $W_2$, but we can always obtain convergence with respect to a different optimal transport cost, namely, the Bregman transport cost $\mathcal{D}_V$ associated with $V$. This is a consequence of Corollary 1 [also see CE17, Proposition 1], which asserts that $\mathcal{D}_V(\mu, \pi) \leq \mathcal{D}_{\mathsf{KL}}(\mu \,\|\, \pi)$.

**Remark 6** (Dimension dependence). Ignoring for now the dependence on $\beta'$ (which may also have a dimension dependence depending on the application), the Bregman divergence $\mathcal{D}_\phi(\pi, \mu_0)$ term is typically of size $O(d)$ (see Section 5 for a particular instance of this). Thus, our overall dimension dependence is $O(d^2)$ for the weakly convex case and $O(d)$ for the strongly convex case. Overall, this is a significantly better dependence on the dimension as compared to the previous works [BDMP17, BEL18]; we perform a detailed comparison in Section 5 for a specific setting.

We also remark that mirror descent has classically been used for dimension reduction by changing the geometry of the algorithm from $\ell_2$ to $\ell_1$. We investigate the possibility of doing the same for sampling in Appendix E.2.

---

[1]In this case, $M_\phi = 0$, so the Lipschitz constant $L$ does not enter the final result. In particular, it is not contradictory to assume strong convexity ($\alpha > 0$).

**Remark 7** (Comparison with [HKRC18]). [HKRC18] show that for strictly log-concave targets, there exists a good mirror map $\phi$ for which the pushforward of the target distribution via $\nabla\phi$ enjoys the same guarantees as ordinary Langevin. However, this result is only existential and gives no guidance on how to construct the mirror map. In contrast, our theorems hold for any choice of mirror map which satisfies our assumptions, and provide guidance on how to choose the mirror map. Also, our relative smoothness condition allows for potentials which blow up at the boundary of their domain (i.e. the target distribution vanishes near the boundary of its support), whereas this is forbidden by the assumptions of [HKRC18]. Lastly, our algorithm does not require computing the third derivative of the mirror map, whereas this is required for [HKRC18].

**Remark 8** (Comparison with [ZPFP20]). [ZPFP20] performs an analysis of the Euler-Maruyama discretization of MLD, which we temporarily refer to as $\mathrm{MLA}'$. Our result guarantees that for any desired accuracy $\epsilon$, it is possible to choose the step size sufficiently small so that MLA achieves the target accuracy; in contrast, the result of [ZPFP20] only guarantees that $\mathrm{MLA}'$ contracts to within a ball around $\pi$ of radius $O(\sqrt{d})$ (measured w.r.t. a modified Wasserstein distance).[2] Moreover, [ZPFP20] conjecture that their bias term is unavoidable.

In light of our result, we believe that it is an interesting open question to resolve their conjecture. Their conjecture, if true, suggests that replacing MLA:2 in our algorithm by a single step of the Euler-Maruyama discretization has disastrous effects on the convergence of the algorithm, and therefore provides further support for considering MLA instead of $\mathrm{MLA}'$.

Recently, after the first draft of our paper was published online, Li et al. [LTVW21] gave an analysis of $\mathrm{MLA}'$ under a subset of the assumptions of [ZPFP20] which indeed exhibits vanishing bias, provided that the relative strong convexity parameter of the potential is sufficiently large compared to the modified self-concordance parameter of the mirror map. It remains an open question to remove this latter restriction from their work, and moreover to obtain similar results under the more usual definitions of relative convexity/smoothness and self-concordance that we adopt in this work.

In order to generalize the discretization analysis from the vanilla Langevin algorithm to the mirror-Langevin algorithm, in the next section we prove a new *displacement convexity* result for the entropy with respect to the Bregman transport cost which may be of independent interest.

## 4 Convexity of the entropy with respect to the Bregman divergence

It is well-known that the entropy functional $\mathcal{H}$ is displacement convex along $W_2$ geodesics [AGS08, Theorem 9.4.11]. In fact, this displacement convexity is crucial in showing that $\mathcal{D}_{\mathsf{KL}}(\cdot \parallel \pi) = \mathcal{E} + \mathcal{H}$ is displacement convex (when $\pi$ is log-concave), which in turn is used to analyze the convergence of LD to the target measure. Therefore, in order to understand the convergence of MLD, it is crucial to see if such a result is true when $W_2$ is replaced by $\mathcal{D}_\phi$. We prove that indeed the displacement convexity-like property holds for $\mathcal{H}$ under $\mathcal{D}_\phi$-optimal couplings.

**Theorem 4** ("Convexity" of the entropy with respect to the Bregman divergence). *Let $\mu$, $\nu$ be probability measures on $\mathbb{R}^d$ and let $X \sim \mu$, $Y \sim \nu$ be coupled according to the Bregman transport cost $\mathcal{D}_\phi(\mu, \nu)$. Then, it holds that*

$$\mathcal{H}(\nu) \geq \mathcal{H}(\mu) + \mathbb{E}\big\langle [\nabla_{W_2}\mathcal{H}(\mu)](X), Y - X \big\rangle.$$

As a corollary, we can use the calculus of optimal transport in order to recover the transportation-cost inequality of [CE17]. In the following, we do not carry out the approximation arguments necessary to make the proof fully correct because a rigorous proof of the statement is already given in [CE17].[3] Rather, our main purpose in giving this argument is simply to point out the convexity principle which underlies the transport inequality.

**Corollary 1** ([CE17, Proposition 1]). *For any probability measure $\mu$ on $\mathbb{R}^d$,*

$$\mathcal{D}_{\mathsf{KL}}(\mu \parallel \pi) \geq \mathcal{D}_V(\mu, \pi).$$

*Proof Sketch.* Let $(X, Y)$ be optimally coupled according to the Bregman transport cost $\mathcal{D}_V(\cdot, \cdot)$ between $\mu$ and $\pi$. We decompose $\mathcal{D}_{\mathsf{KL}}(\mu \parallel \pi) = \mathbb{E}\, V(X) + \mathcal{H}(\mu)$. On one hand, the first term is

$$\mathbb{E}\, V(X) = \mathbb{E}\, V(Y) + \mathbb{E}\langle \nabla V(Y), X - Y \rangle + \mathbb{E}\, D_V(X, Y).$$

---

[2]Notably, the radius of this ball is comparable to the distance at initialization.
[3]In fact, the proof of [CE17] does not even require convexity of $V$.

On the other hand, the convexity result (Theorem 4) shows that

$$\mathcal{H}(\mu) \geq \mathcal{H}(\pi) + \mathbb{E}\langle\nabla_{W_2}\mathcal{H}(\pi)(Y), X - Y\rangle.$$

Putting these together, we obtain

$$\begin{aligned}
\mathcal{D}_{\mathsf{KL}}(\mu \,\|\, \pi) & \\
&\geq \mathbb{E}\, V(Y) + \mathbb{E}\langle\nabla V(Y), X - Y\rangle + \mathbb{E}\, D_V(X, Y) + \mathcal{H}(\pi) + \mathbb{E}\langle\nabla_{W_2}\mathcal{H}(\pi)(Y), X - Y\rangle \\
&= \mathcal{D}_{\mathsf{KL}}(\pi \,\|\, \pi) + \mathbb{E}\big\langle[\nabla V + \nabla_{W_2}\mathcal{H}(\pi)](Y), X - Y\big\rangle + \mathbb{E}\, D_V(X, Y) = \mathbb{E}\, D_V(X, Y),
\end{aligned}$$

since $\nabla V + \nabla_{W_2}\mathcal{H}(\pi)$, the $W_2$ gradient of $\mathcal{D}_{\mathsf{KL}}(\cdot \,\|\, \pi)$ at $\pi$, is zero. This proves the result. $\qquad\square$

# 5  Application to Bayesian logistic regression

In this section, we apply our main result to Bayesian logistic regression. For more applications of our result, such as the possibility of dimension reduction and sampling from non-smooth distributions, we refer the readers to Appendix E.

We recall the setting of Bayesian logistic regression: we observe pairs $(X_i, Y_i)$, $i = 1, \ldots, n$, where $X_i \in \mathbb{R}^d$ and $Y_i \in \{0, 1\}$. The data are assumed to follow the model

$$Y_i \sim \mathrm{Bernoulli}\Big(\frac{\exp\langle\theta, X_i\rangle}{1 + \exp\langle\theta, X_i\rangle}\Big), \qquad \text{independently for } i = 1, \ldots, n. \tag{5.1}$$

Here, the parameter $\theta$ itself is assumed to be a random variable taking values in $\mathbb{R}^d$. If we assume that $\theta$ has a prior density $\lambda$ with respect to Lebesgue measure, then the posterior distribution is

$$\pi(\theta) \propto \lambda(\theta) \exp\Big[\sum_{i=1}^{n}\big(Y_i\,\langle\theta, X_i\rangle - \ln(1 + \exp\langle\theta, X_i\rangle)\big)\Big].$$

Since it may be computationally infeasible to explicitly compute the normalizing constant for the posterior distribution, we turn towards sampling algorithms.

When we take a prior $\lambda$ which has full support on $\mathbb{R}^d$, e.g. a Gaussian prior, then we may apply off-the-shelf methods such as the Langevin diffusion (LD). However, if we choose a prior which has compact support, then the unadjusted Langevin algorithm is no longer an acceptable option because it outputs samples outside the support of the posterior. In this case, we must turn to other methods, such as the projected Langevin algorithm [BEL18]. Here, we explore the use of the mirror-Langevin algorithm (MLA) for constrained sampling.

For the rest of this section, we will focus on a particular problem for concreteness and interpretability: we consider the uniform prior $\lambda$ on the $\ell_\infty$ ball $[-1, 1]^d$. By duality, this is an attractive model when the data $(X_i)_{i=1}^{n}$ have small $\ell_1$-norm, i.e., are approximately sparse. A natural choice of mirror map for this problem is the logarithmic barrier

$$\phi(\theta) = \sum_{i=1}^{d}\Big(\ln\frac{1}{1 - \theta[i]} + \ln\frac{1}{1 + \theta[i]}\Big),$$

where we use $\theta[\cdot]$ to denote the coordinates of $\theta \in \mathbb{R}^d$. Then, $\phi$ is 1-self-concordant ([Nes18, §5.1.3]). We remark that the separability of the mirror map in this example implies that the diffusion step MLA:2 can be simulated in $O(d)$ steps, rather than $O(d^2)$.

For this setting, we compare the guarantees of MLA with the Projected Langevin Algorithm (PLA) [BEL18] and the Moreau-Yosida unadjusted Langevin algorithm (MYULA) [BDMP17]; see Table 1. The details of the comparison are given in Appendix E.1.

We also perform a numerical experiment to compare the practical performance of MLA with PLA. We take $\theta^\star := (0.9, \ldots, 0.9) \in \mathbb{R}^{10}$ as the ground truth, and we generate 1000 i.i.d. pairs $(X_i, Y_i)$ where $X_i$ is sampled uniformly from the $\ell_1$ ball and $Y_i$ is generated from $X_i$ according to (5.1) with $\theta = \theta^\star$. We generate 30 samples using both MLA and PLA (both with step size $\eta = 0.005$). At each iteration, we average the samples to obtain an estimate $\theta_k$ for the posterior mean, and we plot the error $\|\theta_k - \theta^\star\|_2$ in Figure 2, averaged over 10 trials. We implement MLA:2 by performing 10 inner iterations of an Euler-Maruyama discretization.

| Algorithm | Guarantee |
|-----------|-----------|
| MLA | $O(d/\epsilon^2)$ |
| PLA | $O(d^{15}/\epsilon^6)$ |
| MYULA | $O(d^9/\epsilon^3)$ |

Table 1: Comparison of MLA with other constrained sampling algorithms for the number of iterations required to output a sample whose squared total variation distance to $\pi$ is at most $\epsilon$. For simplicity, we focus on the dependence with respect to dimension and the accuracy $\epsilon$, and we defer details of the comparison to Appendix E.1.

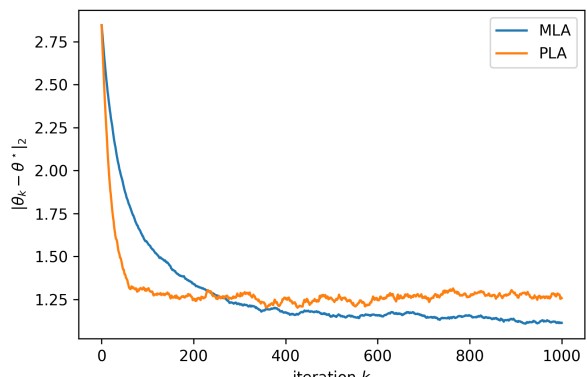

Figure 2: We plot the error for estimators of the posterior mean computed using both MLA and PLA, each taken with step size $\eta = 0.005$.

**Acknowledgments**. Kwangjun Ahn was supported by graduate assistantship from the NSF Grant (CAREER: 1846088) and by Kwanjeong Educational Foundation. Sinho Chewi was supported by the Department of Defense (DoD) through the National Defense Science & Engineering Graduate Fellowship (NDSEG) Program. We thank Philippe Rigollet for helpful suggestions which greatly improved the presentation of this paper.

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
