*Proof.* See Appendix C.2. $\qquad\qquad\qquad\qquad\qquad\qquad\qquad\qquad\qquad\qquad\qquad\qquad\square$

**Theorem 2** (Strongly relatively convex case). *Suppose that Assumptions 1, 2, 3 hold with $\alpha, \beta' > 0$.*

1. *(Convergence in Bregman transport cost) For a target accuracy $\epsilon > 0$, let $X_k \sim \mu_k$ denote the iterates of MLA with step size $\eta = \min\{\frac{\alpha\epsilon}{2\beta'd}, \frac{1}{\beta'}\}$. Then,*

$$\mathcal{D}_\phi(\pi, \mu_N) \leq \epsilon, \quad \text{provided that } N \geq \frac{2\beta'd}{\alpha^2\epsilon}\ln\Big(\frac{2\mathcal{D}_\phi(\pi, \mu_0)}{\epsilon}\Big)\max\Big\{1, \frac{\alpha\epsilon}{2d}\Big\}.$$

2. *(Convergence in KL divergence) For a target accuracy $\epsilon > 0$, suppose that $X_0 \sim \mu_0$ satisfies $\mathcal{D}_\phi(\pi, \mu_0) \leq \epsilon/\alpha$. Let $X_k \sim \mu_k$ denote the iterates of MLA with step size $\eta = \min\{\frac{\alpha\epsilon}{2\beta'd}, \frac{1}{\beta'}\}$. Then, the following convergence rate holds for the mixture distribution $\overline{\mu}_N := \frac{1}{N}\sum_{k=1}^{N} \mu_k$,*

$$\mathcal{D}_{\mathsf{KL}}(\overline{\mu}_N \,\|\, \pi) \leq \epsilon, \quad \text{provided that } N \geq \frac{4\beta'd}{\alpha\epsilon}\max\Big\{1, \frac{\epsilon}{2d}\Big\}.$$

*Proof.* See Appendix C.3. □

Note that the initialization assumption $\mathcal{D}_\phi(\pi, \mu_0) \le \epsilon/\alpha$ in the second assertion of Theorem 2 can be obtained from the first guarantee of Theorem 2. Chaining together the two parts of the theorem, we therefore obtain the following guarantee: suppose that we initialize MLA at a distribution $\mu_0$. Then, with step size $\eta = \min\{\frac{\alpha\epsilon}{2\beta'd}, \frac{1}{\beta'}\}$, we obtain

$$\mathcal{D}_{\mathsf{KL}}\Big(\frac{1}{N_1}\sum_{k=N_0+1}^{N_0+N_1} \mu_k \,\Big\|\, \pi\Big) \le \epsilon, \qquad \text{provided that } \begin{cases} N_0 \ge \frac{2\beta'd}{\alpha\epsilon}\ln\big(\frac{2\alpha\mathcal{D}_\phi(\pi,\mu_0)}{\epsilon}\big)\max\big\{1, \frac{\epsilon}{2d}\big\}, \\ N_1 \ge \frac{4\beta'd}{\alpha\epsilon}\max\big\{1, \frac{\epsilon}{2d}\big\}. \

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

## A  Open questions

We conclude by discussing some questions for future research.

1. As we discuss in Remark 8, it is an open question to determine if the analyses of [ZPFP20, LTVW21] can be improved to obtain vanishing bias for the Euler-Maruyama discretization of MLD under weaker assumptions.

2. In our work, we analyze the sampling analogue of mirror descent under the assumption that the mirror map is self-concordant. This notably bears resemblance to the development of interior-point methodology in optimization [NN94], and it is an interesting problem to develop further sampling analogues of interior-point algorithms.

3. In Appendix E.2, we conduct a preliminary investigation into the possibility that MLA can alleviate the dependence on dimension for some sampling problems. However, Metropolis-adjusted variants of the Langevin algorithm enjoy significantly better dependence on the dimension as compared to their unadjusted counterparts; see [RR98, PST12, CLA$^+$21]. Thus, the Metropolis-adjusted version of MLA may be a more appropriate setting in which to investigate this dimension reduction question, which we leave to future work.

## B  Details on relative convexity and smoothness

For the reader's convenience, we list basic facts regarding relative convexity and smoothness.

**Proposition 1** ([LFN18, Proposition 1.1]). *The following conditions are equivalent:*

- *$f$ is $\beta$-smooth relative to $h$.*

- *$\beta h - f$ is convex on $Q$.*

- *Under twice differentiability, $\nabla^2 f(x) \preceq \beta \nabla^2 h(x)$ for any $x \in \mathrm{int}(Q)$.*

- *$\langle \nabla f(x) - \nabla f(y), x - y \rangle \leq \beta \langle \nabla h(x) - \nabla h(y), x - y \rangle$ for all $x, y \in \mathrm{int}(Q)$.*

*Furthermore, the following conditions are equivalent:*

- *$f$ is $\alpha$-convex relative to $h$.*

- *$f - \alpha h$ is convex on $Q$.*

- *Under twice differentiability, $\nabla^2 f(x) \succeq \alpha \nabla^2 h(x)$ for any $x \in \mathrm{int}(Q)$.*

- *$\langle \nabla f(x) - \nabla f(y), x - y \rangle \geq \alpha \langle \nabla h(x) - \nabla h(y), x - y \rangle$ for all $x, y \in \mathrm{int}(Q)$.*

## C  Proof of the convergence rates

### C.1  Per-iteration progress bound

For the convergence rates of MLA, we first prove the following per-iterate progress bound, from which Theorems 1 and 2 will be easily deduced.

**Lemma 1** (Per-iteration progress bound). *Assume $\beta > 0$. For $0 \leq \eta \leq \frac{1}{\beta}$, let $X_k \sim \mu_k$ be the iterates of* MLA *with step size $\eta$. Then, under Assumptions 1-3, the following holds:*

$$\eta \, \mathcal{D}_{\mathsf{KL}}(\mu_{k+1} \parallel \pi) \leq (1 - \alpha\eta) \, \mathcal{D}_\phi(\pi, \mu_k) - \mathcal{D}_\phi(\pi, \mu_{k+1}) + (\beta + 2M_\phi L)d\eta^2 \,. \qquad \text{(C.1)}$$

*Proof.* We decompose the KL divergence into two parts:

$$\mathcal{D}_{\mathsf{KL}}(\mu \parallel \pi) = \underbrace{\int_Q V(x) \, \mathrm{d}\mu(x)}_{=: \mathcal{E}(\mu)} + \underbrace{\int_Q \mu(x) \ln \mu(x) \, \mathrm{d}x}_{=: \mathcal{H}(\mu)} \,.$$

Here and throughout the paper, we abuse notation by identifying a measure $\mu$ with its density.

The first term above has the interpretation of *energy*, while the second term has the interpretation of (negative) *entropy*. The basic scheme of the proof follows the method in [DMM19], which views the two steps of the update rule MLA as alternately dissipating the energy and the entropy. More specifically, we will show that MLA:1 dissipates $\mathcal{E}$ and MLA:2 dissipates $\mathcal{H}$, while the two steps do not badly interfere with each other.

Our analysis proceeds by controlling each term in the following decomposition:

$$
\begin{aligned}
\mathcal{D}_{\mathsf{KL}}(\mu_{k+1} \,\|\, \pi) &= \mathcal{E}(\mu_{k+1}) + \mathcal{H}(\mu_{k+1}) - \mathcal{E}(\pi) - \mathcal{H}(\pi) \\
&= \underbrace{\mathcal{E}(\mu_{k+1/2}) - \mathcal{E}(\pi)}_{\text{\textcircled{1}}} + \underbrace{\mathcal{E}(\mu_{k+1}) - \mathcal{E}(\mu_{k+1/2})}_{\text{\textcircled{2}}} + \underbrace{\mathcal{H}(\mu_{k+1}) - \mathcal{H}(\pi)}_{\text{\textcircled{3}}}\,.
\end{aligned}
$$

Before we go into the analysis of each term, we outline our proof strategy. Term ① corresponds to a deterministic step of the mirror descent algorithm, and we adapt the analysis of mirror descent based on the Bregman proximal inequality [CT93, Lemma 3.2].

For terms ② and ③, it will be important to understand the stochastic process $(Z_t)_{t\in[0,\eta]}$ in MLA, where $Z_t := \nabla \phi^\star(W_t)$, along with the corresponding marginal laws $(\nu_t)_{t\in[0,\eta]}$. There are two important and distinct perspectives we can adopt. On one hand, the stochastic process $(Z_t)_{t\in[0,\eta]}$ is a diffusion, and can be studied via stochastic calculus. On the other hand, the laws $(\nu_t)_{t\in[0,\eta]}$ follow a Wasserstein "mirror flow" of the entropy functional $\mathcal{H}$, in the sense that it evolves continuously in Wasserstein space with tangent vector $-[\nabla^2\phi]^{-1}\nabla_{W_2}\mathcal{H}(\nu_t)$ (see Section 2.1 for a brief introduction to Wasserstein calculus, and [CLGL$^+$20] for a discussion of MLD from this perspective). In turn, these two perspectives offer different calculation rules: stochastic calculus provides Itô's formula (see [LG16, Theorem 5.10] or [Str18, §3.3]), while Wasserstein calculus provides the rule

$$
\frac{\mathrm{d}}{\mathrm{d}t}\mathcal{F}(\nu_t) = -\mathbb{E}\langle \nabla_{W_2}\mathcal{F}(\nu_t)(Z_t), [\nabla^2\phi(Z_t)]^{-1}\nabla_{W_2}\mathcal{H}(\nu_t)(Z_t)\rangle,
$$

for any sufficiently well-behaved functional $\mathcal{F}$ on Wasserstein space. Both of these perspectives are insightful, and we will employ both.

For term ②, we show that MLA:2 does not greatly increase the energy, and we accomplish this via calculations using Itô's formula together with the relative smoothness and self-concordance assumptions. Finally, we control term ③ by developing a new displacement convexity result (Theorem 4) for the entropy functional $\mathcal{H}$, which is crucial for applying Wasserstein calculus.

①: Let $Y$ be a random variable (defined on the same probability space) which is distributed according to $\pi$. Then,

$$
\begin{aligned}
\mathcal{E}(\mu_{k+1/2}) - \mathcal{E}(\pi) &= \mathbb{E}[V(X_{k+1/2})] - \mathbb{E}[V(Y)] \\
&= \mathbb{E}[V(X_{k+1/2})] - \mathbb{E}[V(X_k)] + \mathbb{E}[V(X_k)] - \mathbb{E}[V(Y)] \\
&\leq \mathbb{E}[\langle \nabla V(X_k), X_{k+1/2} - X_k\rangle + \beta D_\phi(X_{k+1/2}, X_k)] \\
&\quad + \mathbb{E}[\langle \nabla V(X_k), X_k - Y\rangle - \alpha D_\phi(Y, X_k)] \\
&= \mathbb{E}[\langle \nabla V(X_k), X_{k+1/2} - Y\rangle + \beta D_\phi(X_{k+1/2}, X_k) - \alpha D_\phi(Y, X_k)]\,,
\end{aligned}
$$
(C.2)

where the inequality follows due to the $\alpha$-relative strong convexity and $\beta$-relative smoothness of $V$. Now to control (C.2), we invoke a standard tool from optimization:

**Lemma 2** (Bregman proximal inequality [CT93, Lemma 3.2]). *For a convex function $f$ and a convex function $\phi$ of Legendre type, suppose that*

$$
x_+ := \operatorname*{arg\,min}_{z\in\mathcal{Q}}\left[f(z) + D_\phi(z, x)\right].
$$

*Then,*

$$
f(x_+) - f(y) \leq D_\phi(y, x) - D_\phi(y, x_+) - D_\phi(x_+, x) \qquad \forall y \in \mathcal{Q}\,.
$$

Applying the Bregman proximal inequality (Lemma 2) with $f(x) = \eta \langle \nabla V(X_k), x \rangle$,

$$(\text{C.2}) \leq \mathbb{E}\Big[\big(\frac{1}{\eta} - \alpha\big) D_\phi(Y, X_k) - \frac{1}{\eta} D_\phi(Y, X_{k+1/2}) + \big(\beta - \frac{1}{\eta}\big) D_\phi(X_{k+1/2}, X_k)\Big]$$

$$\leq \mathbb{E}\Big[\big(\frac{1}{\eta} - \alpha\big) D_\phi(Y, X_k) - \frac{1}{\eta} D_\phi(Y, X_{k+1/2})\Big],$$

provided that $\frac{1}{\eta} \geq \beta \Leftrightarrow \eta \leq \beta^{-1}$. Choosing $Y$ so that the coupling $(Y, X_k)$ minimizes $\mathbb{E}[D_\phi(Y, X_k)]$, we obtain

$$\eta\left\{\mathcal{E}(\mu_{k+1/2}) - \mathcal{E}(\pi)\right\} \leq (1 - \alpha\eta)\, \mathcal{D}_\phi(\pi, \mu_k) - \mathbb{E}[D_\phi(Y, X_{k+1/2})]$$

$$\leq (1 - \alpha\eta)\, \mathcal{D}_\phi(\pi, \mu_k) - \mathcal{D}_\phi(\pi, \mu_{k+1/2}).$$

②: First, note from MLA that

$$\mathcal{E}(\mu_{k+1}) - \mathcal{E}(\mu_{k+1/2}) = \mathbb{E}\big[V\big(\nabla\phi^\star(W_\eta)\big) - V\big(\nabla\phi^\star(W_0)\big)\big].$$

To compute the above term, we define $f(x) := V(\nabla\phi^\star(x))$ and apply Itô's formula to the random variable $f(W_\eta) - f(W_0)$. To that end, we first compute the Hessian of $f$:

$$\nabla f(x) = \nabla V(\nabla\phi^\star(x))^\top \nabla^2\phi^\star(x) = \nabla V(\nabla\phi^\star(x))^\top [\nabla^2\phi(\nabla\phi^\star(x))]^{-1},$$

$$\nabla^2 f(x) = \nabla^2 V(\nabla\phi^\star(x)) [\nabla^2\phi(\nabla\phi^\star(x))]^{-1} [\nabla^2\phi^\star(x)]$$

$$+ \nabla V(\nabla\phi^\star(x))^\top [\nabla^2\phi(\nabla\phi^\star(x))]^{-1} [\nabla^3\phi(\nabla\phi^\star(x))] [\nabla^2\phi(\nabla\phi^\star(x))]^{-2}.$$

Itô's formula now decomposes $f(W_\eta) - f(W_0)$ into the sum of an integral and a stochastic integral. Intuitively, the stochastic integral has mean zero (since it is a local martingale), and this can be rigorously argued using the standard technique of localization; we give the argument at the end of this step. Thus, we concentrate on the expectation of the first term. Writing $Z_t := \nabla\phi^\star(W_t)$, the above Hessian calculation gives

$$\mathbb{E}[f(W_\eta) - f(W_0)] \tag{C.3}$$

$$= \mathbb{E}\Big[\int_0^\eta \big\langle \nabla^2 V(Z_t) [\nabla^2\phi(Z_t)]^{-2}, \nabla^2\phi(Z_t)\big\rangle \, \mathrm{d}t\Big]$$

$$+ \mathbb{E}\Big[\int_0^\eta \big\langle \nabla V(Z_t)^\top [\nabla^2\phi(Z_t)]^{-1} [\nabla^3\phi(Z_t)] [\nabla^2\phi(Z_t)]^{-2}, \nabla^2\phi(Z_t)\big\rangle \, \mathrm{d}t\Big]$$

$$= \mathbb{E}\Big[\int_0^\eta \big\langle \nabla^2 V(Z_t), [\nabla^2\phi(Z_t)]^{-1}\big\rangle \, \mathrm{d}t\Big] \tag{C.4}$$

$$+ \mathbb{E}\Big[\int_0^\eta \mathrm{tr}\big(\nabla V(Z_t)^\top [\nabla^2\phi(Z_t)]^{-1} [\nabla^3\phi(Z_t)] [\nabla^2\phi(Z_t)]^{-1}\big) \, \mathrm{d}t\Big].$$

$$\tag{C.5}$$

We can control (C.4) easily based on the relative smoothness of $V$: indeed, since $\nabla^2 V \preceq \beta\nabla^2\phi$ (see Appendix B),

$$(\text{C.4}) \leq \beta d\eta.$$

To control (C.5), we use the self-concordance of $\phi$. We recall here the following result:

**Proposition 2** ([Nes18, Corollary 5.1.1]). *A function $\phi$ is self-concordant with a constant $M_\phi \geq 0$ if and only if for any $x \in \mathrm{dom}(\phi)$ and any direction $u \in \mathbb{R}^n$ we have*

$$\nabla^3\phi(x)u \preceq 2M_\phi \|u\|_{\nabla^2\phi(x)} \nabla^2\phi(x).$$

Using Proposition 2, it follows that

$$(\text{C.5}) \leq 2M_\phi \int_0^\eta \mathbb{E}\big[\big\|[\nabla^2\phi(Z_t)]^{-1}\nabla V(Z_t)\big\|_{\nabla^2\phi(Z_t)} \mathrm{tr}\big([\nabla^2\phi(Z_t)] [\nabla^2\phi(Z_t)]^{-1}\big)\big] \, \mathrm{d}t$$

$$\leq 2M_\phi d \int_0^\eta \mathbb{E}\big[\|\nabla V(Z_t)\|_{[\nabla^2\phi(Z_t)]^{-1}}\big] \, \mathrm{d}t \leq 2M_\phi L d\eta.$$

Thus, our calculation shows that

$$\mathcal{E}(\mu_{k+1}) - \mathcal{E}(\mu_{k+1/2}) \leq (\beta + 2M_\phi L)d\eta. \tag{C.6}$$

We now sketch the localization argument. Let $(\tau_\ell)_{\ell \in \mathbb{N}}$ be a localizing sequence for $(W_t)_{t \in [0,\eta]}$. The argument above may be applied rigorously for the stopped process $(W_{t \wedge \tau_\ell})_{t \in [0,\eta]}$ to obtain $\mathbb{E}\, V(Z_{\eta \wedge \tau_\ell}) - \mathbb{E}\, V(Z_0) \leq (\beta + 2M_\phi L)d\eta$. Since $V$ is bounded below, we use Fatou's lemma to pass $\ell \to \infty$ and deduce (C.6).

③: Let $\nu_t$ denote the law of $Z_t := \nabla\phi^\star(W_t)$. For this step, we start with a calculation of the derivative of $t \mapsto \mathcal{D}_\phi(\pi, \nu_t)$. Noting that $\nabla_2 D_\phi(y, x) = -\nabla^2\phi(x)\,(y - x)$ and that $\nu_t$ follows the Wasserstein tangent vector $-[\nabla^2\phi]^{-1}\nabla_{W_2}\mathcal{H}(\nu_t)$, we expect that

$$\frac{\mathrm{d}}{\mathrm{d}t}\mathcal{D}_\phi(\pi, \nu_t) = \mathbb{E}\langle [\nabla^2\phi(Z_t)]^{-1}\nabla_{W_2}\mathcal{H}(\nu_t)(Z_t), \nabla^2\phi(Z_t)\,(Y - Z_t)\rangle$$
$$= \mathbb{E}\langle \nabla_{W_2}\mathcal{H}(\nu_t)(Z_t), Y - Z_t\rangle,$$

where $(Y, Z_t)$ are optimally coupled for $\pi$ and $\nu_t$ for the Bregman transport cost. In general, the differentiability properties of optimal transport costs can be quite subtle, but thankfully it is much easier to establish the superdifferentiability

$$\frac{\mathrm{d}}{\mathrm{d}t^+}\mathcal{D}_\phi(\pi, \nu_t) \leq \mathbb{E}\langle \nabla_{W_2}\mathcal{H}(\nu_t)(Z_t), Y - Z_t\rangle$$

at almost all $t$, which is all that will be needed for the subsequent argument. The superdifferentiability result is proven along the lines of [OV00, Lemma 2]; see also [AGS08, Theorem 10.2.2] or the proof of [Vil09, Theorem 23.9].

Next, we apply a result which can be interpreted as convexity of the entropy functional with respect to the Bregman divergence; it will be given as Theorem 4 in the next section. It implies that for $t \in [0, \eta]$,

$$\frac{\mathrm{d}}{\mathrm{d}t^+}\mathcal{D}_\phi(\pi, \nu_t) \leq \mathbb{E}\langle \nabla_{W_2}\mathcal{H}(\nu_t)(Z_t), Y - Z_t\rangle \leq \mathcal{H}(\pi) - \mathcal{H}(\nu_t) \leq \mathcal{H}(\pi) - \mathcal{H}(\nu_\eta),$$

where the last inequality follows since

$$\frac{\mathrm{d}}{\mathrm{d}t}\mathcal{H}(\nu_t) = -\mathbb{E}\big[\langle \nabla_{W_2}\mathcal{H}(\nu_t)(Z_t), [\nabla^2\phi(Z_t)]^{-1}\nabla_{W_2}\mathcal{H}(\nu_t)(Z_t)\rangle\big] \leq 0,$$

which implies $\mathcal{H}(\nu_\eta) \leq \mathcal{H}(\nu_t)$ for any $t \in [0, \eta]$. Integrating from $0$ to $\eta$, we obtain

$$\mathcal{D}_\phi(\pi, \nu_\eta) - \mathcal{D}_\phi(\pi, \nu_0) \leq \eta\left\{\mathcal{H}(\pi) - \mathcal{H}(\nu_\eta)\right\},$$

which is the same as

$$\eta\left\{\mathcal{H}(\mu_{k+1}) - \mathcal{H}(\pi)\right\} \leq \mathcal{D}_\phi(\pi, \mu_{k+1/2}) - \mathcal{D}_\phi(\pi, \mu_{k+1}).$$

Combining the upper bounds from ①, ②, and ③, the proof of Lemma 1 is complete. □

## C.2 Proof of Theorem 1

From the per-iteration progress bound (Lemma 1), we have for any $k \in \mathbb{N}$

$$\eta\,\mathcal{D}_{\mathsf{KL}}(\mu_{k+1} \,\|\, \pi) \leq \mathcal{D}_\phi(\pi, \mu_k) - \mathcal{D}_\phi(\pi, \mu_{k+1}) + \beta' d\eta^2. \tag{C.7}$$

Summing (C.7) over $k = 0, 1, \ldots, N - 1$,

$$\eta\sum_{k=1}^{N}\mathcal{D}_{\mathsf{KL}}(\mu_k \,\|\, \pi) \leq \mathcal{D}_\phi(\pi, \mu_0) - \mathcal{D}_\phi(\pi, \mu_N) + \beta' d\eta^2 N.$$

Using the convexity of the KL divergence [which follows from the Gibbs variational principle; see RAS15, §5.1],

$$\mathcal{D}_{\mathsf{KL}}(\bar{\mu}_N \,\|\, \pi) \leq \frac{\mathcal{D}_\phi(\pi, \mu_0)}{N\eta} + \beta' d\eta \leq \frac{\epsilon}{2} + \frac{\epsilon}{2},$$

where the last inequality follows from the choice $N \geq \frac{2\mathcal{D}_\phi(\pi, \mu_0)}{\eta\epsilon}$ and $\eta \leq \frac{\epsilon}{2\beta d}$.

## C.3 Proof of Theorem 2

Let us first prove the convergence in Bregman transport cost. For any $k \in \mathbb{N}$, the per-iteration progress bound (Lemma 1) together with the fact $\mathcal{D}_{\mathsf{KL}}(\mu_{k+1} \| \pi) \geq 0$ imply

$$\mathcal{D}_\phi(\pi, \mu_{k+1}) \leq (1 - \alpha\eta) \, \mathcal{D}_\phi(\pi, \mu_k) + \beta' d\eta^2 \,. \tag{C.8}$$

Recursively applying (C.8) for $k = 0, 1, \ldots, N-1$, we obtain

$$\mathcal{D}_\phi(\pi, \mu_N) \leq (1 - \alpha\eta)^N \, \mathcal{D}_\phi(\pi, \mu_0) + \beta' d\eta^2 \sum_{k=0}^{N-1} (1 - \alpha\eta)^k$$

$$\leq (1 - \alpha\eta)^N \, \mathcal{D}_\phi(\pi, \mu_0) + \beta' d\eta^2 \sum_{k=0}^{\infty} (1 - \alpha\eta)^k$$

$$\leq \exp(-\alpha\eta N) \, \mathcal{D}_\phi(\pi, \mu_0) + \frac{\beta' d\eta}{\alpha} \leq \frac{\epsilon}{2} + \frac{\epsilon}{2} \,,$$

where the last inequality follows since $N \geq \frac{1}{\alpha\eta} \ln \frac{2\mathcal{D}_\phi(\pi, \mu_0)}{\epsilon}$ and $\eta \leq \frac{\alpha\epsilon}{2\beta'd}$. Having proved the convergence in terms of the Bregman transport cost, the convergence in terms of the KL divergence follows by applying Theorem 1.

## C.4 Analysis for the non-smooth case (Theorem 3)

The analysis for the non-smooth case proceeds in a similar manner to the smooth case. We first prove the following per-iterate progress bound.

**Lemma 3** (Per-iteration progress bound; non-smooth case). *Let $X_k \sim \mu_k$ be the iterates of* MLA *with step size $\eta > 0$. Assume that $\phi$ is 1-strongly convex w.r.t $\|\cdot\|$, and that $V$ is convex and $\tilde{L}$-Lipschitz w.r.t $\|\cdot\|$. Then, the following holds:*

$$\eta \, \mathcal{D}_{\mathsf{KL}}(\mu_{k+1} \| \pi) \leq \mathcal{D}_\phi(\pi, \mu_{k+1/2}) - \mathcal{D}_\phi(\pi, \mu_{k+3/2}) + \frac{\eta^2 \tilde{L}^2}{2} \,. \tag{C.9}$$

*Proof.* Our analysis proceeds by controlling each term in the following decomposition:

$$\mathcal{D}_{\mathsf{KL}}(\mu_{k+1} \| \pi) = \underbrace{\mathcal{E}(\mu_{k+1}) - \mathcal{E}(\pi)}_{\text{(A)}} + \underbrace{\mathcal{H}(\mu_{k+1}) - \mathcal{H}(\pi)}_{\text{(B)}} \,.$$

For term (B), we invoke the following upper bound (from the analysis of term ③ in the proof of Lemma 1):

$$\eta \{\mathcal{H}(\mu_{k+1}) - \mathcal{H}(\pi)\} \leq \mathcal{D}_\phi(\pi, \mu_{k+1/2}) - \mathcal{D}_\phi(\pi, \mu_{k+1}) \,. \tag{C.10}$$

Let us turn to (A), and let $Y$ be a random variable (defined on the same probability space) which is distributed according to $\pi$. Since we have

$$X_{k+3/2} = \underset{x \in \mathcal{Q}}{\arg\min} \left[ \langle \eta \nabla V(X_{k+1}), x \rangle + D_\phi(x, X_{k+1}) \right] \,,$$

applying the Bregman proximal inequality (Lemma 2) with $f(x) = \eta \langle \nabla V(X_{k+1}), x \rangle$ gives

$$\eta \langle \nabla V(X_{k+1}), X_{k+3/2} - Y \rangle \leq D_\phi(Y, X_{k+1}) - D_\phi(Y, X_{k+3/2}) - D_\phi(X_{k+3/2}, X_{k+1}) \,,$$

which after rearranging becomes

$$D_\phi(Y, X_{k+3/2}) - D_\phi(Y, X_{k+1}) \leq \eta \langle \nabla V(X_{k+1}), Y - X_{k+3/2} \rangle - D_\phi(X_{k+3/2}, X_{k+1}) \,. \tag{C.11}$$

On the other hand, the right hand side of (C.11) can be controlled using the convexity, the Lipschitz-ness of $V$, and strong convexity of $\phi$:

RHS of (C.11)

$$= \eta \langle \nabla V(X_{k+1}), Y - X_{k+1} \rangle + \eta \langle \nabla V(X_{k+1}), X_{k+1} - X_{k+3/2} \rangle - D_\phi(X_{k+3/2}, X_{k+1})$$

$$\leq \eta \left[ V(Y) - V(X_{k+1}) \right] + \eta \|\nabla V(X_{k+1})\|_\star \|X_{k+1} - X_{k+3/2}\| - \frac{1}{2} \|X_{k+1} - X_{k+3/2}\|^2$$

$$\leq \eta \left[ V(Y) - V(X_{k+1}) \right] + \frac{\eta^2}{2} \|\nabla V(X_{k+1})\|_\star^2$$

$$\leq \eta \left[ V(Y) - V(X_{k+1}) \right] + \frac{\eta^2 \tilde{L}^2}{2} \,.$$

For the LHS of (C.11), choose $Y$ so that the coupling $(Y, X_{k+1})$ minimizes $\mathbb{E}[D_\phi(Y, X_{k+1})]$ to obtain

$$\mathbb{E}[\text{LHS of (C.11)}] = \mathbb{E}[D_\phi(Y, X_{k+3/2})] - \mathcal{D}_\phi(\pi, \mu_{k+1}) \geq \mathcal{D}_\phi(\pi, \mu_{k+3/2}) - \mathcal{D}_\phi(\pi, \mu_{k+1}).$$

Combining these upper and lower bounds, (C.11) becomes:

$$\eta\left[\mathcal{E}(\mu_{k+1}) - \mathcal{E}(\pi)\right] \leq \mathcal{D}_\phi(\pi, \mu_{k+1}) - \mathcal{D}_\phi(\pi, \mu_{k+3/2}) + \frac{\eta^2 \tilde{L}^2}{2}.$$

Together with (C.10), the proof is complete. $\qquad\square$

Now using Lemma 3, we prove Theorem 3.

*Proof of Theorem 3.* From Lemma 3, we have for any $k \in \mathbb{N}$

$$\eta\, \mathcal{D}_{\mathsf{KL}}(\mu_{k+1} \,\|\, \pi) \leq \mathcal{D}_\phi(\pi, \mu_{k+1/2}) - \mathcal{D}_\phi(\pi, \mu_{k+3/2}) + \frac{\eta^2 \tilde{L}^2}{2}. \qquad\text{(C.12)}$$

Summing (C.12) over $k = 0, 1, \ldots, N-1$,

$$\eta \sum_{k=1}^{N} \mathcal{D}_{\mathsf{KL}}(\mu_k \,\|\, \pi) \leq \mathcal{D}_\phi(\pi, \mu_{1/2}) - \mathcal{D}_\phi(\pi, \mu_{N+1/2}) + \frac{\eta^2 \tilde{L}^2}{2}\, N.$$

Again using the convexity of the KL divergence, we obtain

$$\mathcal{D}_{\mathsf{KL}}(\overline{\mu}_N \,\|\, \pi) \leq \frac{\mathcal{D}_\phi(\pi, \mu_{1/2})}{N\eta} + \frac{\eta \tilde{L}^2}{2} \leq \frac{\epsilon}{2} + \frac{\epsilon}{2},$$

where the last inequality follows from the choice $N \geq \frac{2\mathcal{D}_\phi(\pi, \mu_{1/2})}{\eta\epsilon}$ and $\eta \leq \frac{\epsilon}{\tilde{L}^2}$. $\qquad\square$

## D    Proofs for the convexity of entropy

To prove Theorem 4, we will use the known result about the convexity of $\mathcal{H}$ along generalized geodesics [AGS08, Theorem 9.4.11]. To that end, the first step is to obtain a characterization of the optimal Bregman transport coupling which is analogous to Brenier's theorem. The following theorem is of independent interest:

**Theorem 5** (Brenier's theorem for the Bregman transport cost)**.** *Let $\mu$, $\nu$ be probability measures on $\mathbb{R}^d$. The optimal Bregman transport coupling $(X, Y)$ for $\mu$ and $\nu$ is of the form*

$$\nabla\phi(X) - \nabla\phi(Y) = \nabla h(X),$$

*where $h : \mathbb{R}^d \to \mathbb{R} \cup \{-\infty\}$ is such that $\phi - h$ is convex..*

*Proof.* From the general theory of optimal transport duality, it holds that

$$\nabla_1 D_\phi(X, Y) = \nabla h(X),$$

where $h$ is a $D_\phi$-concave function [see Vil09, Theorem 10.28].[4] The left-hand side of this equation evaluates to $\nabla\phi(X) - \nabla\phi(Y)$, so we simply have to check that $D_\phi$-concavity of $h$ implies that $\phi - h$ is convex (which is in fact equivalent to saying that $h$ is 1-relatively smooth with respect to $\phi$, see Appendix B).

Recall that the $D_\phi$-concavity of $h$ means there exists a function $\tilde{h} : \mathbb{R}^d \to \mathbb{R} \cup \{-\infty\}$ such that

$$h(x) = \inf_{y \in \mathbb{R}^d}\{D_\phi(x, y) - \tilde{h}(y)\},$$

---

[4]In fact, there are three assumptions for [Vil09, Theorem 10.28]. Here, we explicitly check them one by one for clarity. (i) *Super-differentiability:* $D_\phi$ is clearly differentiable on $\mathcal{Q}$ as $\phi$ is of class $C^3$. (ii) *Injectivity of gradient:* $\nabla_1 D_\phi(x, \cdot) = \nabla\phi(x) - \nabla\phi(\cdot)$ is injective as $\phi$ is of Legendre type. (iii) *$\mu$-almost-sure differentiability of $D_\phi$-concave functions:* In (D.1), we actually show that for any $D_\phi$-concave function $h$, $\phi - h$ is convex and thus differentiable Lebesgue a.e. [Roc70, Theorem 25.5]. Since $\mu$ is absolutely continuous w.r.t. Lebesgue measure and $\phi$ is differentiable, $h$ must be differentiable $\mu$-almost surely.

see [Vil09, Definition 5.2].[5] If we expand out the definition of the Bregman divergence, we can rewrite this as

$$\phi(x) - h(x) = \sup_{y \in \mathbb{R}^d} \{\langle \nabla \phi(y), x - y \rangle + \tilde{h}(y) + \phi(y)\}. \tag{D.1}$$

As a supremum of affine functions, we see that $\phi - h$ is convex, which completes the proof. $\qquad\square$

We now prove Theorem 4 using Theorem 5:

*Proof of Theorem 4.* By Theorem 5, the optimal Bregman transport coupling is of the form $\nabla \phi(Y) = \nabla(\phi - h)(X) = \nabla \zeta(X)$, where we have defined the convex function $\zeta := \phi - h$. Hence, letting $\overline{\nu}$ denote the law of $\overline{Y} := \nabla \phi(Y)$, it follows that $(X, \nabla \zeta(X))$ is a $W_2$ optimal coupling between $\mu$ and $\overline{\nu}$. Furthermore, since $\phi$ is a convex function of Legendre type, $(\nabla \zeta(X), \nabla \phi^\star \circ \nabla \zeta(X))$ is also a $W_2$ optimal coupling between $\overline{\nu}$ and $\nu$. Noting that

$$\nabla \phi^\star \circ \nabla \zeta(X) = \nabla \phi^\star \circ \nabla \phi(Y) = Y,$$

it follows that $(X, Y)$ is a generalized geodesic according to $W_2$. Therefore, the convexity of $\mathcal{H}$ along generalized geodesics [AGS08, Theorem 9.4.11] concludes the proof (see [SKL20, Lemma 4]). $\qquad\square$

**Remark 9.** For reader's convenience, we provide a direct calculation that (formally) shows the convexity result. Since we have shown $Y = \nabla \phi^\star \circ \nabla \zeta(X)$, the change of variable formula gives

$$\mathcal{H}(\nu) = \int \nu(y) \ln \nu(y) \, dy = \int \mu(x) \ln \nu \big( \nabla \phi^\star \circ \nabla \zeta(x) \big) \, dx$$

$$= \int \mu(x) \ln \frac{\mu(x)}{\det \nabla (\nabla \phi^\star \circ \nabla \zeta)(x)} \, dx.$$

(Here the change of variables is valid since $[\nabla(\nabla \phi^\star \circ \nabla \zeta)](x) = [\nabla^2 \phi^\star(\nabla \zeta(x))] \, [\nabla^2 \zeta(x)] \succeq 0$.) Thus, using the convexity of $-\ln \det$ and integrating by parts, we obtain

$$\mathcal{H}(\mu) - \mathcal{H}(\nu) = -\int \mu(x) \ln \det \nabla(\nabla \phi^\star \circ \nabla \zeta)(x) \, dx$$

$$\geq -\int \mu(x) \operatorname{tr}[\nabla(\nabla \phi^\star \circ \nabla \zeta)(x) - I_d] \, dx = \int \langle \nabla \mu(x), (\nabla \phi^\star \circ \nabla \zeta)(x) - x \rangle \, dx$$

$$= \int \langle \nabla \ln \mu(x), (\nabla \phi^\star \circ \nabla \zeta)(x) - x \rangle \, d\mu(x).$$

Recalling that $\nabla_{W_2} \mathcal{H}(\mu) = \nabla \ln \mu$ and $Y = \nabla \phi^\star(\nabla \zeta(X))$, the convexity result follows.

# E  Further applications

## E.1  Details for the Bayesian logistic regression application

We may compute

$$V(\theta) = \sum_{i=1}^n \big( -Y_i \langle \theta, X_i \rangle + \ln(1 + \exp \langle \theta, X_i \rangle) \big), \qquad \phi(\theta) = \sum_{i=1}^d \Big( \ln \frac{1}{1 - \theta[i]} + \ln \frac{1}{1 + \theta[i]} \Big)$$

$$\nabla V(\theta) = -\sum_{i=1}^n \Big( Y_i - \frac{\exp \langle \theta, X_i \rangle}{1 + \exp \langle \theta, X_i \rangle} \Big) X_i, \qquad \nabla \phi(\theta) = \sum_{i=1}^d \Big( \frac{1}{1 - \theta[i]} - \frac{1}{1 + \theta[i]} \Big) e_i,$$

$$\nabla^2 V(\theta) = \sum_{i=1}^n \frac{\exp \langle \theta, X_i \rangle}{(1 + \exp \langle \theta, X_i \rangle)^2} X_i X_i^\top, \qquad \nabla^2 \phi(\theta) = \operatorname{diag}\Big[ \frac{1}{(1 - \theta)^2} + \frac{1}{(1 + \theta)^2} \Big].$$

---

[5]In Villani's book, he works with the definition of *c-convexity* rather than *c-concavity*, but this is merely a matter of convention; c.f. [Vil03, §2.4] for the conventions regarding *c*-concavity.

From these expressions, we see that

$$0 \preceq \nabla^2 V \preceq \sum_{i=1}^{n} X_i X_i^\top, \qquad 2I_d \preceq \nabla^2 \phi.$$

Let $L := \sup_{[-1,1]^d} \|\nabla V\|$ denote the (ordinary) Lipschitz constant of $V$, and let $\beta$ denote the (ordinary) smoothness parameter of $V$ (from above we see that $\beta$ can be taken to be the largest eigenvalue of $\sum_{i=1}^{n} X_i X_i^\top$). Note that the 2-strong convexity of $\phi$ implies that $V$ is $L/\sqrt{2}$-relatively Lipschitz and $\beta/2$-relatively smooth with respect to $\phi$, so Theorem 1 holds with

$$\beta' = \frac{\beta}{2} + \sqrt{2}L.$$

In order to fully understand the quantitative convergence rate provided by Theorem 1, we must also bound the Bregman divergence $\mathcal{D}_\phi(\pi, \mu_0)$. We have:

**Lemma 4.** *Let $\mu_0 = \delta_0$ be the point mass at $0$. Then, for the logarithmic barrier mirror map $\phi$ defined above, we have $\mathcal{D}_\phi(\pi, \mu_0) \leq 4.1 \, (1 + \beta + L) \, d$.*

*Proof.* See Appendix F. $\qquad\qquad\square$

From Theorem 1, we can deduce that using $N$ iterations of MLA, we can obtain a distribution $\mu_N^{\mathsf{MLA}}$ such that

$$2\|\mu_N^{\mathsf{MLA}} - \pi\|_{\mathrm{TV}}^2 \leq \mathcal{D}_{\mathsf{KL}}(\mu_N^{\mathsf{MLA}} \,\|\, \pi) \leq \epsilon, \qquad \text{provided } N \geq \frac{23(1 + \beta + L)^2 d^2}{\epsilon^2} \max\left\{1, \frac{\epsilon}{2d}\right\},$$

where $\beta$ is the largest eigenvalue of $\sum_{i=1}^{n} X_i X_i^\top$ and $L := \sup_{[-1,1]^d} \|\nabla V\|$ is the usual Lipschitz constant of $V$; for details, see Appendix E.1. In fact, if we use the non-smooth guarantee in Theorem 3, then we can improve this to $O(L^2 d/\epsilon^2)$ iterations. For comparison purposes, the guarantee for the projected Langevin algorithm (PLA) [BEL18, Theorem 1] with $R = \sqrt{d}$ implies

$$\|\mu_N^{\mathsf{PLA}} - \pi\|_{\mathrm{TV}}^2 \leq \epsilon, \qquad \text{provided } N \geq \widetilde{\Omega}\Big(\frac{(\sqrt{d} + \beta + L)^{12} d^9}{\epsilon^6}\Big).$$

On the other hand, the Moreau-Yosida unadjusted Langevin algorithm (MYULA) [BDMP17, Theorem 2] with $R = \sqrt{d}$ provides the guarantee[6]

$$\|\mu_N^{\mathsf{MYULA}} - \pi\|_{\mathrm{TV}}^2 \leq \epsilon, \qquad \text{provided } N \geq \widetilde{\Omega}\Big(\frac{\beta^4 d^9}{\epsilon^3}\Big).$$

### E.2  Better dimension dependency via mirror Langevin

As described in [Bub15, §4.3], a classical application of mirror descent is to obtain better dependence on the dimension by changing the geometry of the optimization algorithm from $\ell_2$ to $\ell_1$. We investigate the possibility of analogous improvements in the setting of constrained sampling.

We consider a simple toy problem in which the constraint set is the interior of the filled-in simplex $\mathcal{Q} := \{x \in \mathbb{R}^d \mid x > 0, \sum_{i=1}^{d} x[i] < 1\}$, and we take the potential to be a quadratic

$$V(x) := \frac{1}{2}\langle x, Ax \rangle,$$

where $A \in \mathbb{R}^{d \times d}$ is a symmetric positive semidefinite matrix with all entries bounded in magnitude by 1. We choose as our mirror map the barrier:

$$\phi(x) := \sum_{i=1}^{d} \ln \frac{1}{x[i]} + \ln \frac{1}{1 - \sum_{i=1}^{d} x[i]}.$$

---

[6]To be precise, their bound on the number of iterates required reads $N \geq \widetilde{\Omega}(\Delta_2^4 d^7/\epsilon^3)$, where $\Delta_2$ is a parameter measuring how close the domain $\mathrm{dom}(V)$ is to an isotropic convex body. For concreteness, we bound this parameter by $\beta R$ following [BDMP17, pg. 7].

This map is self-concordant with parameter 1.

We can compute

$$\nabla \phi(x) = \sum_{i=1}^{d} \left( -\frac{1}{x[i]} + \frac{x[i]}{1 - \sum_{j=1}^{d} x[j]} \right) e_i,$$

$$\nabla^2 \phi(x) = \operatorname{diag} \frac{1}{x^2} + \frac{I_d}{1 - \sum_{i=1}^{d} x[i]} + \frac{xx^\top}{\left(1 - \sum_{i=1}^{d} x[i]\right)^2}.$$

Since $x[i] < 1$ for all $i = 1, \ldots, d$, it follows that $\langle v, \operatorname{diag}(1/x^2)v \rangle \geq \langle v, \operatorname{diag}(1/x)v \rangle \geq \|v\|_1^2$, where the second inequality follows from the strong convexity of the entropy with respect to the $\ell_1$-norm. Hence, $\phi$ is 1-strongly convex with respect to the $\ell_1$-norm. From our assumption on $A$,

$$\|\nabla V(x)\|_{[\nabla^2 \phi(x)]^{-1}} \leq \|\nabla V(x)\|_\infty \leq 1,$$

$$\langle v, \nabla^2 V(x)v \rangle \leq \left| \sum_{i,j=1}^{d} A_{i,j} v_i v_j \right| \leq \sum_{i,j=1}^{d} |v_i||v_j| \leq \|v\|_1^2,$$

which implies that $V$ is 1-relatively Lipschitz and 1-relatively smooth with respect to $\phi$, and the assumptions of Theorem 1 hold with $\beta' = 3$. In contrast, if we had instead considered the $\ell_2$-norm, then the Lipschitz constant of $V$ could be as large as $\sqrt{d}$, and the smoothness parameter of $V$ could be as large as $d$. Together with a warm start, this suggests that MLA could attain a better dimension dependence for this example.

**Remark 10.** Alternatively, we can apply Theorem 3 with the entropic mirror map

$$\phi(x) = \sum_{i=1}^{d} x[i] \ln x[i] + \left(1 - \sum_{i=1}^{d} x[i]\right) \ln\left(1 - \sum_{i=1}^{d} x[i]\right)$$

to the above setting; note that Theorem 3 only requires standard assumptions for mirror descent guarantees (e.g., [Bub15, Theorem 4.2]), and does not require the mirror map to be self-concordant. In particular, $V$ is 1-Lipschitz w.r.t. $\|\cdot\|_1$ and $\phi$ is strongly convex w.r.t. $\|\cdot\|_1$, so Theorem 3 implies that $\mathcal{D}_{\mathsf{KL}}(\overline{\mu}_N \parallel \pi) \leq \epsilon$ after $N = O\left(\frac{\mathcal{D}_\phi(\pi, \mu_{1/2})}{\epsilon^2}\right)$ iterations. For comparison, note that the approach of Hsieh et al. [HKRC18] does not apply to this example, because the pushforward of the distribution via the entropic mirror map is not log-concave.

In Figure 3, we report the results of a preliminary numerical study which indicates that MLA may have an advantage in high dimension. In this example, we generate a $100 \times 100$ matrix $\widetilde{A}$ with i.i.d. entries drawn uniformly from $[-1, 1]$, and we take $A$ to be the matrix $\widetilde{A}\widetilde{A}^\top$, rescaled so the largest magnitude of the entries is 1. We compare the performance of MLA (with step size $\eta = 5 \times 10^{-3}$ and 10 inner iterations of the Euler-Maruyama discretization for MLA:2) with PLA (taken with step size $\eta = 10^{-6}$). The step sizes were tuned to be as large as possible while avoiding instabilities in the algorithms. We plot the convergence in $W_2^2$, averaged over 10 trials.

### E.3 Sampling from non-smooth distributions

Thus far, we have focused on distributions whose potential $V$ is bounded within its domain $\operatorname{dom}(V)$. However, in many applications, one is required to sample from a distribution whose potential $V$ blows up near the boundary of its domain. Such distributions violate the standard assumptions of Lipschitzness and smoothness and hence are beyond the scope of the existing guarantees. In this subsection, we demonstrate that one can still sample from such distributions via MLA together with the relative Lipschitzness and relative smoothness.

Consider the Dirichlet distribution $\pi$ which is defined on the interior of the filled-in simplex $\mathcal{Q} := \{x \in \mathbb{R}^d \mid x > 0, \sum_{i=1}^{d} x[i] < 1\}$ by the potential

$$V(x) = a_0 \ln \frac{1}{1 - \sum_{i=1}^{d} x[i]} + \sum_{i=1}^{d} a_i \ln \frac{1}{x[i]},$$

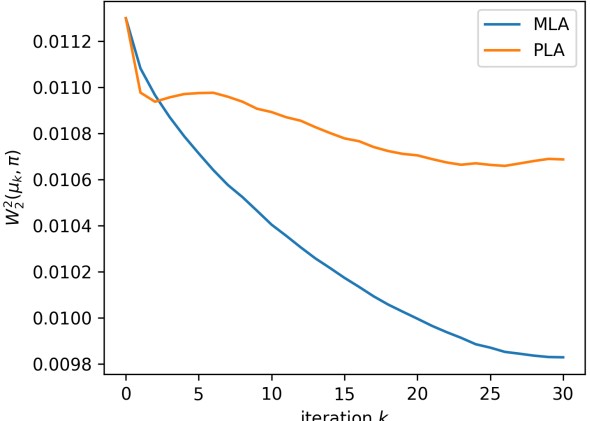

Figure 3: 30 iterations of MLA vs. PLA for a 100-dimensional example with step size $5 \times 10^{-3}$ for MLA and $10^{-6}$ for PLA. The step sizes were tuned to be as large as possible while avoiding instability. The results are averaged over 10 runs to smooth out the curves.

for some constants $a_0, a_1, \ldots, a_d > 0$, and we take $V = \phi$. Then, it is well-known that $V$ is $(\max_{i=0,1,\ldots,d} a_i^{-1/2})$-self-concordant [Nes18, Theorem 5.1.1]. Also, from $(\sum_{i=0}^{d} a_i)$-exp-concavity of $V$ [Nes18, Theorem 5.3.2], it holds that $\|\nabla V(x)\|_{[\nabla^2 V(x)]^{-1}} \leq (\sum_{i=0}^{d} a_i)^{1/2}$. Therefore, it follows that $V$ is $(\sum_{i=0}^{d} a_i)^{1/2}$-Lipschitz, $1$-convex, and $1$-smooth relative to $V$, which implies that the assumptions of Theorem 2 holds with

$$\beta' = 1 + 2 \left( \max_{i=0,1,\ldots,d} a_i^{-1/2} \right) \left( \sum_{i=0}^{d} a_i \right)^{1/2} \leq 3\sqrt{d} \sqrt{\frac{a_{\max}}{a_{\min}}},$$

where $a_{\max} := \max_{i=0,1,\ldots,d} a_i$ and $a_{\max} := \min_{i=0,1,\ldots,d} a_i$. Therefore, one can obtain a mixture distribution $\overline{\mu}_N$ after $N$ iterations of MLA such that

$$\mathcal{D}_{\mathsf{KL}}(\overline{\mu}_N \,\|\, \pi) \leq \epsilon, \quad \text{provided that } N \geq \widetilde{\Omega}\left( \sqrt{\frac{a_{\max}}{a_{\min}}} \frac{d^{3/2}}{\epsilon} \right).$$

Using this example, we perform a numerical experiment to investigate the effect of simulating the diffusion step (MLA:2) more faithfully. In Figure 4, we run MLA for 30 iterations with step size 0.005. The potential $V$ and the mirror map $\phi$ are taken as above, with $a_0 = a_1 = \cdots = a_{10} = 2$. When implementing MLA:2, we use $k$ inner iterations of an Euler-Maruyama discretization, where $k$ ranges in $\{1, 5, 10, 20\}$, and the results are averaged over 10 trials. We do indeed observe that a more precise implementation of MLA:2 yields better results, although the difference is subtle.

## F    Auxiliary results

**Lemma 5.** *Let $\pi$ be a probability distribution supported on $[-1, 1]^d$ which has density proportional to $\exp(-V)$. Assume that $V : [-1, 1]^d \to \mathbb{R}^d$ is $L$-Lipschitz and $\beta$-smooth. Then, we have the following bound on the marginal density $\pi_1$ of $\pi$ on the first coordinate:*

$$\sup_{[-1,1]} \pi_1 \leq 3(1 + \sqrt{\beta} + L).$$

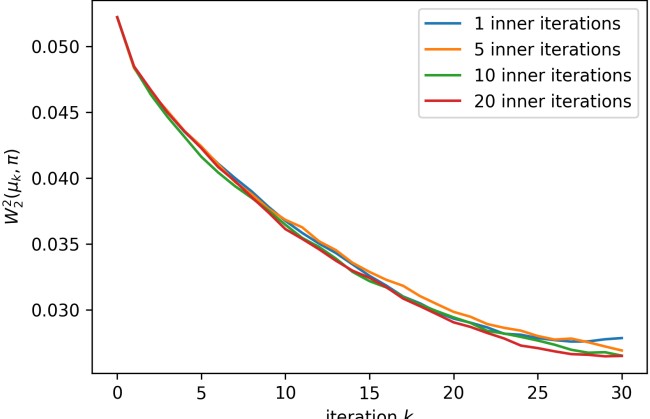

Figure 4: We investigate the effect of implementing MLA:2 using 1, 5, 10, or 20 steps of an Euler-Maruyama discretization. A more faithful implementation of MLA:2 appears to bring slight benefits.

*Proof.* Let $Z := \int_{[-1,1]^d} \exp(-V)$ denote the normalizing constant, let $\theta_1^\star \in [-1,1]$ be the maximizer of $\pi_1$, and let $\theta \in [-1,1]^d$.[7] We can write $\theta = (\theta_1, \theta_{-1})$, where $\theta_{-1} \in \mathbb{R}^{d-1}$.[7] Then,

$$V(\theta) \le V(\theta_1^\star, \theta_{-1}) + \partial_1 V(\theta_1^\star, \theta_{-1})(\theta_1 - \theta_1^\star) + \frac{\beta}{2}(\theta_1 - \theta_1^\star)^2$$

$$\le V(\theta_1^\star, \theta_{-1}) + L|\theta_1 - \theta_1^\star| + \frac{\beta}{2}(\theta_1 - \theta_1^\star)^2$$

$$\le \frac{1}{2} + V(\theta_1^\star, \theta_{-1}) + \frac{\beta + L^2}{2}(\theta_1 - \theta_1^\star)^2.$$

This yields the lower bound

$$\pi_1(\theta_1) = \frac{1}{Z}\int_{[-1,1]^{d-1}} \exp\big(-V(\theta_1, \theta_{-1})\big)\, d\theta_{-1}$$

$$\ge \frac{\exp[-(\beta + L^2)(\theta_1 - \theta_1^\star)^2/2 - 1/2]}{Z}\int_{[-1,1]^{d-1}} \exp\big(-V(\theta_1^\star, \theta_{-1})\big)\, d\theta_{-1}$$

$$= \exp\left[-\frac{1}{2}(\beta + L^2)(\theta_1 - \theta^\star)^2 - \frac{1}{2}\right]\sup_{[-1,1]}\pi_1.$$

Next,

$$1 = \int_{[-1,1]} \pi_1(\theta_1)\, d\theta_1 \ge \frac{\sup_{[-1,1]}\pi_1}{\sqrt{e}}\int_{[-1,1]} \exp\left[-\frac{1}{2}(\beta + L^2)(\theta_1 - \theta^\star)^2\right] d\theta_1$$

$$\ge \frac{\sup_{[-1,1]}\pi_1}{\sqrt{e}}\int_0^1 \exp\left[-\frac{1}{2}(\beta + L^2)x^2\right] dx.$$

Let $c := \int_0^1 \exp(-x^2)\, dx$. By splitting into the two cases $\beta + L^2 \le 1$ and $\beta + L^2 \ge 1$, we can deduce the inequality

$$1 \ge \frac{c\sup_{[-1,1]}\pi_1}{\sqrt{e}}\left(\frac{1}{\sqrt{\beta + L^2}} \wedge 1\right).$$

It yields

$$\sup_{[-1,1]}\pi_1 \le \frac{\sqrt{e}}{c}\left(\sqrt{\beta + L^2} \vee 1\right) \le \frac{\sqrt{e}}{c}\left((\sqrt{\beta} + L) \vee 1\right) \le \frac{\sqrt{e}}{c}\left(1 + \sqrt{\beta} + L\right),$$

which is the result. □

---

[7]In Section 5 we used the notation $\theta[i]$ for the $i$th coordinate of $\theta$, but for the sake of simplicity we switch to the notation $\theta_i$ for this proof.

*Proof of Lemma 4.* Let $\pi_i$ denote the $i$th marginal of $\pi$. Then, since $\phi(0) = \nabla\phi(0) = 0$, we must estimate

$$\mathcal{D}_\phi(\pi, \mu_0) = \int_{[-1,1]^d} \sum_{i=1}^d \ln \frac{1}{1 - \theta[i]^2} \, \pi(\theta) \, d\theta = \sum_{i=1}^d \int_{[-1,1]} \ln \frac{1}{1 - \theta[i]^2} \, \pi_i(\theta[i]) \, d\theta[i]$$

$$\leq C \left(1 + \sqrt{\beta} + L\right) d \int_{[-1,1]} \ln \frac{1}{1 - x^2} \, dx \leq \frac{3}{2} C \left(1 + \beta + L\right) d \int_{[-1,1]} \ln \frac{1}{1 - x^2} \, dx,$$

where $C$ is the constant from the proof of Lemma 5. It yields the result. $\qquad\square$