# OpenReview forum: "Efficient constrained sampling via the mirror-Langevin algorithm"
_NeurIPS.cc/2021/Conference — NeurIPS 2021 Poster_

### Official Review · Reviewer_FccJ · 2021-07-07

**Rating:** 6
**Confidence:** 4

**Summary:**

This work proposes an analysis of the symmetric Mirror Langevin diffusion, in the case where the diffusion step in the mirror space is computed exactly. Under milder assumptions than existing work [ZPFP20], the authors prove convergence of the error in KL divergence to 0 as opposed to previous work showing convergence to a Wasserstein ball with non-zero radius, hence showing the benefit of better approximating the diffusion step than simply doing an Euler-Maruyama step.

**Limitations And Societal Impact:**

The authors adequately addressed the limitations and potential negative societal impact.

**Main Review:**

Pros:
- Interesting theoretical result, showing that the convergence issue in [ZPFP20] can be solved by a better approximation of the diffusion step.
- Relaxes the assumptions made in [ZPFP20], in particular the commutation conditions for Hessians.

Cons:
- The paper would greatly improve by analyzing the impact of approximate, which would provide a more correct description of the practical convergence rate. In particular, the claim in the abstract that “Unlike prior work, our result has vanishing bias as the step size goes to zero” is not fair, since including a discretization of the diffusion will most probably involve a similar bias.
- The practical usefulness of the algorithm seems limited. Based on Figure 4, it is not clear whether a better diffusion discretization truly yields a better solution. Moreover, the algorithm is only compared to PLA, which has the worst convergence rate, and is known to perform rather badly in practice. It may be interesting to compare against MYULA [BDMP17], MLD (asymmetric discretization [HKRC18], or  [REKV20] (the latter also being worth mentioning in Table 1).

[ZPFP20] Kelvin Shuangjian Zhang, Gabriel Peyré, Jalal Fadili, and Marcelo Pereyra. Wasserstein control of mirror Langevin Monte Carlo

[HKRC18] Ya-Ping Hsieh, Ali Kavis, Paul Rolland, and Volkan Cevher. Mirrored Langevin dynamics.

[BDMP17] Nicolas Brosse, Alain Durmus, Eric Moulines, and Marcelo Pereyra. Sampling from a log-concave distribution with compact support with proximal Langevin Monte Carlo.

[REKV20] Paul Rolland, Armin Eftekhari, Ali Kavis, and Volkan Cevher. Double-loop unadjusted Langevin algorithm.


**Time Spent Reviewing:**

2h

---

> ### Author Response · Authors · 2021-08-07
> **Response**
>
> Thank you for your review. We address your points:
>
> (1) We agree that analyzing the impact of approximate diffusion steps (e.g. via Euler-Maruyama discretization) is a natural next step. For example, it would be interesting to characterize how the error behaves as a function of the number of inner approximation steps. We believe this is an important future direction for research, but out of scope for the present work which mainly aims to study whether the asymptotic bias of previous work is unavoidable. We will clarify our abstract to avoid overstating our claims; thank you for pointing this out.
>
> (2) Regarding simulations, we agree that our experiments are not adequate to evaluate the practical performance of MLA. However, as our main contribution is theoretical in nature, we believe that it is better to leave extensive numerical comparisons for future work. We greatly appreciate the references, and we will add the comparison with [REKV20] in Table 1 as per your suggestion.

---

### Official Review · Reviewer_UqoF · 2021-07-15

**Rating:** 6
**Confidence:** 4

**Summary:**

The paper under review is considering a new discretization for the MLD. The motivation comes from solving constrained optimization problem. Under proper assumption, the authors present convergence analysis for the algorithm in terms of the KL and Bregman divergence.  In order to derive the convergence analysis, the author derive displacement convexity property and treat/verify several technical conditions following previous works in the literature.

**Limitations And Societal Impact:**

Questions and comments.

1.  The condition $N>\frac{2\mathcal D(\pi||d\mu_0)}{\eta\varepsilon}$ in Theorem 1 seems to be big. Is there any guideline how does this help in practice.

2. MLA 1 and MLA 2 seems to be quite general. Can this be adapted to reversible/nonreversible variable diffusion Langevin diffusion? Can you deal with the non-convex $V$ case? For example, the mirror Langevin is one special case of the gradient drift variable Langevin diffusion considered in [Q. Feng and W. Li: Hypoelliptic entropy dissipation for stochastic differential equations ].  The entropy decay and LSI are derived for general Langevin diffusions. ( here diffusion matrix $a(X_t)= \nabla^2\phi(X_t)$ ) Do you think that this may help to derive the convergence analysis for more general potential fiction $V$?



**Main Review:**

Positive.

1. The paper is well written. The preparation of the materials, the main proof outlines, and the organization of the paper are in good manner.

2. The Proof seems to be easy to follow; I did not check all the computations and the all the coefficients in the proof process. The logic and the basic stream seems to be correct. The authors show some subtle points and technique conditions that were used in the proof.

3. The analysis of the discretization algorithms seem to work for more general Langevin diffusions, e.g. variable diffusion Langevin dynamics.

**Time Spent Reviewing:**

8

---

> ### Author Response · Authors · 2021-08-07
> **Response**
>
> Thank you for your review. We address your two main points:
>
> (1) Our convergence guarantees are in line with other convergence results proven in the literature, and in particular, they reduce to the state-of-the-art convergence guarantees for the ordinary Langevin algorithm when the mirror map is the squared Euclidean norm.
>
> (2) The application of MLA for non-convex potentials has already been initiated under the assumption of a “mirror LSI”, see https://nbviewer.jupyter.org/github/QijiaJ/QijiaJ.github.io/blob/master/_includes/mirror_langevin_2021.pdf. We do not know if our techniques can be adopted to analyze this case, nor do we know if they can handle more general diffusion matrices. This is an interesting question which is left for future research.

---

### Official Review · Reviewer_AWLg · 2021-07-15

**Rating:** 7
**Confidence:** 5

**Summary:**

The paper proposes a new mirror descent like Langevin algorithm to sample from exp(-V), where V is relatively smooth/convex.
The algorithm is designed as a Forward Flow algorithm.
The convergence guarantees are similar to those of the vanilla Langevin algorithm (ULA), but using some newly defined Bergman divergence instead of the Wasserstein 2 distance.

**Limitations And Societal Impact:**

Limitations: the authors could discuss more the implementation of MLD2 and acknowledge clearly that the complexity of one iteration of the proposed algorithm **is** higher than one iteration of ULA of [ZPFP20].

**Main Review:**

Originality:

The paper extends the approach of [DMM19] to handle a mirror descent like Langevin algorithm. To this end, the authors define a Bregman optimal transport cost that will induce the natural geometry of the algorithm. Basically, the convergence rate are provided in terms of the Bregman cost if V is relatively strongly convex and in terms of KL else.



The algorithm is designed as a "Forward Flow" algorithm [Wib18], so that the convergence proof of [DMM19] can be extended to mirror descent. The algorithm has two steps. The first step is like a mirror descent descent and the second one introduce randomness following a mirror flow  (instead of a Gaussian noise in ULA).

On the practical side, unlike ULA, this mirror flow step cannot be implemented in closed form. The implementation of MLD2 should be discussed further. Can we implement it in closed form for some \phi (other than the square norm)?

On the theoretical side, this paper is quite rich mathematically with several interesting ideas that extend those of DMM19 and can be used beyond the scope of this paper. Indeed, although there is a direct parallel between the proof of [DMM19] and this paper, the adaptation of [DMM19] is highly non trivial. Basically, every steps is more complex, in more details:

-the step that looks like a gradient descent proof in DMM19 here looks like a mirror descent proof
-the dissipation of the entropy along the mirror flow relies on the **provable** (Theorem 4) convexity of the entropy along geodesics induced by the Bregman optimal transport cost. Inspecting the proof, it is a consequence of the geodesic convexity of the entropy along generalized geodesics.
-the dissipation of energy along the mirror flow relies in DMM19 on a simple smoothness argument and here relies on Ito calculus.


Quality:

The paper is well written. Even the appendix is of high quality. The convergence rates are natural extensions of those of ULA, I just have some concern about the warm start assumption in Th 2 (convergence in KL divergence). Moreover, in the proof of Corollary 1, it is used that the entropy is (sub)differentiable at \pi, and this requires implicitly the density of \pi to be an element of S_{loc}^{1,1}(R^d) (see [AGS08, Th 10.4.13]).

Additionally, I would be happy to see the Forward flow interpretation of the algorithm in the main paper, as well as the Wasserstein mirror flow interpretation of MLD2. In particular, how do you find the formula of the velocity field?

The simulations are a bit lightweight. I understand the this work is theoretical. In Figure 2, why not plotting the ergodic means of the \theta_k along the algorithm? (they are supposed to converge a.s. to the mean a posteriori thanks to the ergodic theorem).

Finally, some cleaning is needed in the references.

Clarity:

The assumptions should be recall in the statement of each theorem. In particular, do the authors assume simultaneously Ass. 2 and Ass. 3 with \alpha > 0 in Theorem 2? It seems that these assumptions are mainly incompatible (think about a Lipschitz strongly convex function for the case where \phi is the square norm)

Significance:

Overall, the paper is a smart extension of DMM19 and, even if one could argue that the practical implementation should be discussed further, the paper brings new ideas on how to use optimization techniques in sampling and that should be relevant to the community.

MINOR:

l.134: FYI, this is an equivalence, see [A, Lemma 2.2.1]
l. 231: "V is relatively convex"
l. 242: "O(d) for the weakly convex case with a warm start" Why? I don't understand.
l. 664: "known" <-- "new"
l. 685: conclusion of the proof is missing. One can conclude using Lemma 4 in [SKL20]
l. 305: Better to write "it output samples outside..."
l. 108: Capital letter missing
l. 314: MLA2
l. 615: superdifferentiability argument: Why is the entropy (sub)differentiable at \nu_t ?


[A]: S. Brazitikos, A. Giannopoulos, P. Valettas, and B.-H. Vritsiou. Geometry of isotropic convex bodies


**Time Spent Reviewing:**

3

---

> ### Author Response · Authors · 2021-08-07
> **Response**
>
> Thank you for your detailed review. We address your points below.
>
> (1) Forward flow: This is an insightful point and this is in fact precisely the perspective taken in a follow-up work (https://nbviewer.jupyter.org/github/QijiaJ/QijiaJ.github.io/blob/master/_includes/mirror_langevin_2021.pdf). The derivation of the vector field can be found in e.g. Appendix A of the linked paper. For clarity, we will mention this interpretation in our paper as per your suggestion.
>
> (2) Closed form for diffusion: We are not aware of mirror maps that yield closed forms for the diffusion, although this seems plausible for simple choices of the mirror map. For simulations, we implemented all steps approximately.
>
> (3) Warm start in Thm. 2: Is the confusion stemming from the second statement in Thm. 2? To be clear, we do not need to assume a warm start w.r.t. the Bregman distance because the first statement of Thm. 2 provides this warm start. In other words, the conclusion of the second statement holds by starting from any initial distribution, with a number of iterations N given by the sum of the two bounds in the two statements.
>
> (4) Subdifferentiability of entropy: You are absolutely correct; in order to make the proof fully rigorous, we would need to supplement it with an approximation argument. However, our main goal in giving this corollary is simply to point out that there is a convexity statement underlying the theorem of Cordero-Erausquin; we do not wish to rigorously reprove the result because, in any case, the proof of Cordero-Erausquin holds under more general assumptions (in fact, Cordero-Erausquin does not even require log-concavity). Thank you for pointing this out; we will clarify the text accordingly.
>
> (5) Simulations: We fully agree that more extensive experiments are needed to better evaluate the practical performance MLA (your suggestion here is a good one). However, as you point out, our primary contribution is theoretical and we believe that it is better to leave further experiments for future work.
>
> (6) Assumptions: This is an excellent point. Our main motivation indeed comes from the constrained setting (the domain is compact) in which case the assumptions are not incompatible. The specific case of the squared norm mirror map is special: in the definition of our parameter β’, since the self-concordance parameter is 0, then the fact that the mirror map is not Lipschitz does not enter our convergence result. We will clarify this as well as add the assumptions of the theorems to the theorem statements for clarity.
>
> (7) Limitations: Thank you for pointing this out, we will edit to reflect accordingly.
>
> (8) Minor technical comments: Thank you for the references and comments. We will address them accordingly.

---

> > ### Comment · Reviewer_AWLg · 2021-08-16
> > **Thanks for the detailed answers**
> >
> > OK, thanks. I will keep my score as it is.
> >
> > (3): Please clarify in the paper. It was not clear at all for me.

---

### Official Review · Reviewer_d6un · 2021-07-19

**Rating:** 5
**Confidence:** 4

**Summary:**

This paper proposes and studies a new discretization scheme of the mirror-Langevin diffusion proposed in Zhang et al. (2020). The results build on the assumption of self-concordance of the mirror function and has vanishing bias as the step size tends to zero.

**Limitations And Societal Impact:**

Yes.

**Main Review:**

Strengths:
* The motivation is strong --- correcting bias in the Euler-Maruyama discretization of Zhang et al.
* The proof looks correct to me. It is also easy to follow and the results are neat.

Weaknesses:
* The biggest concern for me is that the proposed algorithm is not really a discretization since the diffusion step is still in continuous time and the proof relies on Ito's formula. This obviously makes a big difference and it is unclear what a proper discretization of it is. I noticed that this has been the concern for reviewers of COLT. However, I don't think it is now addressed up to satisfactory.
* I wonder how the proposed algorithm compare to running LD in the mirrored space, which was shown in Ya-Ping Hsieh et al. (2018) to have O(d/eps^2) convergence. For optimization this would not be a concern since the optimum need not match under transformation of the mirror map, but since now we are dealing with sampling, there is no obvious reason that one should prefer MLA over LD in the mirrored space.
* The empirical result (Figure 2), if the authors decide to include any, does not provide strong evidence supporting the theoretical argument. PLA seems to converge much faster, although MLA lead to better results but there is still a large gap from the target distribution.
* How strong is the self-concordance assumption on mirror maps? Does it apply to choices other than barriers?

**Time Spent Reviewing:**

4

---

> ### Author Response · Authors · 2021-08-07
> **Response**
>
> Thank you for your review. We will address your points in turn.
>
> (1) Discretization: Theoretically, our work follows the well-established Nemirovsky-Yudin model of computation which measures the complexity of an algorithm via the number of queries it makes to a local oracle (in this case, the oracle is for the gradient of the potential V). Practically, our proposed algorithm can be implemented approximately by discretizing the diffusion step more finely than the gradient step; indeed, one of our main contributions is the surprising message that vanishing bias may require treating the gradient and diffusion steps differently. It is true that our algorithm corresponds to an idealized limit when the diffusion is simulated perfectly, and a more refined analysis is required to quantify how finely the diffusion step should be discretized. However, we believe that this lies beyond the scope of the current work, which is intended to provide a clean analysis and draw attention to this new phenomenon.
>
> We would like to also emphasize that the discretization considered in our work is conceptually useful for understanding the phenomenon we identify, and in fact it has already inspired a successful follow-up study (https://nbviewer.jupyter.org/github/QijiaJ/QijiaJ.github.io/blob/master/_includes/mirror_langevin_2021.pdf) whose conclusions agree with ours.
>
>
> (2) Comparison with mirrored Langevin: Thank you for the question; we will add a discussion of this in our paper. For context, we briefly summarize Hsieh et al.:
>
> - Algorithm: their mirrored Langevin algorithm (denoted MLA’ for this response) corresponds to running the standard Langevin algorithm on the pushforward of the target distribution via the gradient of the mirror map. In particular, without the diffusion, their algorithm does not reduce to mirror descent, whereas ours does.
>
> - Main results: Their results for MLA’ are twofold.
> (i) Their Thm. 3 shows that for strongly log-concave targets, there exists a good mirror map for which MLA’ enjoys the same guarantees as ordinary Langevin. However, this result is only existential and gives no guidance on how to construct it.
> (ii) They also identify one example on the simplex for which they can construct a good mirror map by hand.
>
> We list some ways that our MLA results improve upon MLA’:
>
> 1. Our theorem holds for any choice of mirror map which satisfies our assumptions. Also, our convergence results provide guidance on how to choose the mirror map. This lies in contrast to Thm. 3 of Hsieh et al.
>
> 2. Our result applies under strictly weaker assumptions. We only require log-concavity, not strong log-concavity. Also, our relative smoothness condition allows for potentials which blow up at the boundary of their domain (i.e. the target distribution vanishes near the boundary of its support), whereas this is forbidden by the assumptions of Hsieh et al.
>
> 3.  Our algorithm does not require computing the third derivative of the mirror map, whereas this is required for MLA’.
>
> (3) Empirical results: We agree that further experiments are required to evaluate the practical performance of the algorithm. As our contribution is primarily theoretical in nature, our aim in providing experiments was not to demonstrate practical superiority but rather a simple sanity check.
>
> (4) Self-concordance: We would like to quickly note that our results only need the mirror map to be self-concordant not a self-concordant barrier. Hence, our results hold for various mirror maps that are not log-barriers, see e.g. https://en.wikipedia.org/wiki/Self-concordant_function. Assuming self-concordance without the barrier property is useful, not only to accommodate a wider range of mirror maps, but also because our convergence results when applied to self-concordant barriers do not depend on the barrier parameter (which is typically much larger than the self-concordance parameter).

---

### Decision · Program_Chairs · 2021-09-27

**Decision:**

Accept (Poster)

**Comment:**

The authors have identified an assumption that extends the work of [Zhang, Peyre, Fadili, Pereyra] to have guarantees for the mirrored Langevin algorithm -- not to be confused from the mirrored Langevin dynamics of [Hsieh et al].

However. the assumption cannot be implemented in practice since the authors cannot do the perfect discretization in the flow step. The standard Euler discretization normally results in the Cox-Ingersoll-Ross (CIR) process, which has a notoriously slow convergence rate. While the authors use a different discretization process, it cannot be implemented. The additional assumptions of the relative smoothness + strong convexity seems restrictive.

Even in the case of the simplex constraints, the authors use the log-barrier and not the usual entropy, which is the staple of mirror descent methods. Moreover, there is no comparison with [Hsieh et al] for the same example. Note that those authors use the entropic mirror map.

There are weaknesses as there is no analysis on the proposed discretization and the ensuing bias introduced while the paper is beautifully written with solid mathematical content. In particular, as one of the reviewers point out, this is not a real discrete time paper nor a continuous time paper - but it is somewhere in between. A revised paper with some analysis on the discretization would make it super compelling.